# TableDART: Dynamic Adaptive Multi-Modal Routing for Table Understanding

**Xiaobo Xing[1][*], Wei Yuan[1][*], Tong Chen[1], Quoc Viet Hung Nguyen[2], Xiangliang Zhang[3], Hongzhi Yin[1][†]**

[1]The University of Queensland, Australia
[2]Griffith University, Australia    [3]University of Notre Dame, USA
{ xiaobo.xing, w.yuan, tong.chen, h.yin1 }@uq.edu.au
 henry.nguyen@griffith.edu.au, xzhang33@nd.edu

## Abstract

Modeling semantic and structural information from tabular data remains a core challenge for effective table understanding. Existing Table-as-Text approaches flatten tables for large language models (LLMs), but lose crucial structural cues, while Table-as-Image methods preserve structure yet struggle with precise semantics. Recent Table-as-Multimodality strategies attempt to combine textual and visual views, but they (1) statically process both modalities for every query-table pair within large multimodal LLMs (MLLMs), inevitably introducing redundancy and even conflicts, and (2) depend on costly fine-tuning of MLLMs. In light of this, we propose TableDART, a training-efficient framework that integrates multimodal views by reusing pretrained single-modality models. TableDART introduces a lightweight 2.59M-parameter MLP gating network that dynamically selects the optimal path (Text-only, Image-only, or Fusion) for each table–query pair, reducing redundancy and avoiding conflicts that arise when textual and visual views of the same table provide inconsistent cues. By routing to the most appropriate view, our framework improves both accuracy and efficiency. In addition, we propose a novel agent to mediate cross-modal knowledge integration by analyzing outputs from text- and image-based models, either selecting the best result or synthesizing a new answer through reasoning. This design avoids the prohibitive costs of full MLLM fine-tuning. Extensive experiments on seven benchmarks show that TableDART establishes new state-of-the-art performance among open-source models, surpassing the strongest baseline by an average of 4.02%. The code is available at: https://github.com/xiaobo-xing/TableDART.

## 1 Introduction

Tabular data is one of the most ubiquitous formats in the real world, with applications spanning finance (Chen et al., 2021; Zhu et al., 2021; 2024b), healthcare (Shi et al., 2024; Wang et al., 2024), and many other domains that rely on relational databases and spreadsheets (Fey et al., 2024; Gao et al., 2026). Despite its prevalence, effectively modeling tabular data remains a long-standing challenge, due to its unique characteristics such as heterogeneity of diverse data types across columns, permutation invariance regarding row and column order, and hierarchical structure of multi-level or nested headers and indices (Fang et al., 2024; Borisov et al., 2024; Chen et al., 2023).

Existing approaches to tabular data understanding can be broadly categorized into two paradigms. The first is the "*Table-as-Text*" paradigm, where tables are linearized into sequences and processed by large language models (LLMs) (Zha et al., 2023; Su et al., 2024). While effective, this approach is sensitive to serialization choices (Sui et al., 2024) and inevitably loses critical structural information (Deng et al., 2024). The second paradigm treats "*Table-as-Image*", where table screenshots are processed with vision models. This approach better preserves structural information but struggles to capture precise and aligned semantic meaning (Deng et al., 2024).

---

[*]Co-first authors with equal contributions.    [†]Corresponding author.

To combine the complementary strengths of these paradigms, recent research has moved toward "*Table-as-Multimodality*", where tables are represented in both textual and visual forms and processed via fine-tuned multimodal LLMs (MLLMs) (Liu et al., 2025). While promising, this MLLM-based paradigm suffers from two key limitations that hinder both performance and practicality. First, regardless of the context, they mandate both modality views for every table-query pair. However, not all queries benefit from multiple views, as integrating multiple views often introduces redundancy and, in some cases, conflicting signals. For example, textual linearization may inadvertently impose row-order sensitivity (Fang et al., 2024), while tables in their original or visual form remain permutation-invariant (Wu et al., 2025). Such redundancy and conflicts mislead the MLLM's table understanding, resulting in suboptimal performance. Second, existing approaches typically rely on heavy MLLMs, which remain computationally prohibitive to adapt at scale even with parameter-efficient fine-tuning strategies (Hu et al., 2022), limiting their practicality under constrained training budgets and deployment pipelines (Yin et al., 2025).

In this paper, we propose TableDART (Dynamic Adaptive multi-modal RouTing), a novel framework for efficient and effective multimodal table understanding. Unlike prior work, TableDART does not naively combine all modalities' perspectives. Instead, it introduces a lightweight MLP gating network that dynamically selects the optimal processing path (either Text-only, Image-only, or Fusion) for each table-query pair depending on the instance's complexity and resource efficiency. When fusion is selected, a powerful LLM agent integrates the outputs of both single-modality experts, serving either as an arbitrator (selecting the better output) or as a rescuer (generating an enhanced answer by reasoning over both sources). Notably, TableDART reuses existing single-modality experts (Text-only and Image-only), requiring training only for the gating network. This makes it significantly more training-efficient than fine-tuning MLLMs. We validate TableDART with extensive experiments on seven diverse benchmarks. Results show that TableDART not only achieves state-of-the-art performance among open-source models, surpassing the strongest MLLM-based baseline by an average of 4.02% accuracy, but also provides new insights into dynamic routing policies for multimodal table understanding.

In summary, our main contributions are:

(a) We propose TableDART, a novel framework that adaptively selects the optimal processing path for each query to make the most of modality-specific features for table understanding.

(b) We introduce a parameter-efficient training paradigm that fully leverages pretrained (and frozen) single-modality experts by learning to route over experts with different output vocabularies, making the lightweight gating network (2.59M parameters) the only trainable part. An LLM agent is designed to further facilitate cross-modal consensus, enabling a plug-and-play setup without fine-tuning any LLM or MLLM backbones.

(c) We conduct extensive experiments and establish new state-of-the-art performance on multiple benchmarks. In addition, we provide an in-depth analysis of the learned routing policy, demonstrating its strong generalizability and effectiveness.

## 2 RELATED WORK

**Unimodal Approaches for Table Understanding.** A dominant paradigm for applying LLMs to tabular data is the Table-as-Text approach, which serializes tables into linear text sequences. Early work such as TaPas (Herzig et al., 2020) demonstrated its effectiveness, inspiring specialized tabular models like TableLlama (Zhang et al., 2024) and TableGPT2 (Su et al., 2024), which achieve state-of-the-art results for table understanding. However, these models remain constrained by their input modality, as they cannot capture the rich visual and structural semantics inherent in the original table format (Deng et al., 2024). To overcome this limitation, the Table-as-Image paradigm has emerged, leveraging Multi-modal Large Language Models (MLLMs) to directly process table images. Pioneering works such as Table-LLaVA (Zheng et al., 2024), TabPedia (Zhao et al., 2024), SynTab-LLaVA (Zhou et al., 2025), and generalist MLLMs like Ovis2 (Lu et al., 2024) and Qwen2.5-VL (Bai et al., 2025) showcase the potential of modeling visual layouts and structural cues, aligning with broader trends in vision-centric text understanding (Yuan et al., 2026; Qiao et al., 2026). Yet, this paradigm is not a universal solution, as its effectiveness diminishes for tasks where reasoning relies heavily on text-centric pretraining knowledge (Deng et al., 2024).

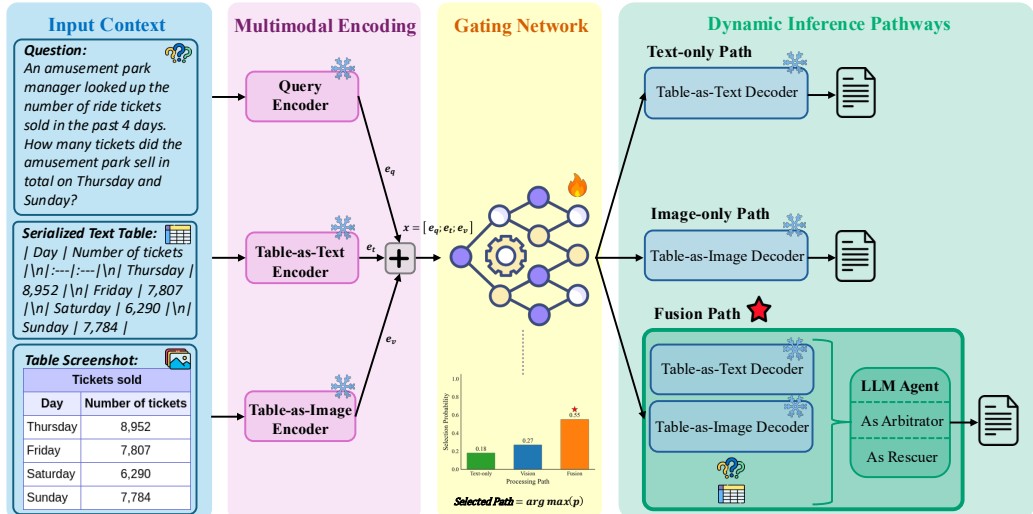

Figure 1: **Architecture of TableDART.** The framework operates in three main stages: Multimodal Encoding (Section 3.2), Gating Network (Section 3.3), and Dynamic Inference Pathways (Section 3.4).

**Multi-modal Approaches and Dynamic Routing.** Building on the complementary strengths of unimodal representations, Table-as-Multimodality approaches have emerged. For example, HIPPO (Liu et al., 2025) fine-tunes an MLLM to jointly process both text and image representations of the table, achieving strong performance gains. However, these MLLM-based methods typically adopt a static, one-size-fits-all strategy, applying both modalities to every table–query pair. This rigid design can introduce redundancy or even conflict, particularly when a single modality would suffice. To address this gap, we propose TableDART, which introduces a dynamic routing mechanism that adaptively selects the optimal modality path for each instance. By learning an intelligent, instance-level policy, TableDART enables more efficient and robust multimodal reasoning, moving beyond the limitations of prior paradigms.

# 3 METHODOLOGY

## 3.1 OVERVIEW

The key novelty of TableDART lies in its ability to dynamically and efficiently leverage existing single-modality models to capture complementary information from tabular data for downstream tasks. To this end, TableDART integrates five components: (1) a Table-as-Text model $\mathcal{M}_t$, (2) a Table-as-Image model $\mathcal{M}_v$, (3) a text embedding model to process queries, (4) a lightweight gating network, and (5) an LLM-based agent responsible for multimodal fusion. Among these components, only the gating network is trainable, ensuring that TableDART remains highly parameter- and training-efficient. Figure 1 provides an overview of TableDART's workflow, which is detailed in the subsequent sections.

## 3.2 ENCODING TABLE-AS-MULTIMODALITY

Given a raw table $\mathcal{T}$ and a query $q$, TableDART leverages existing models to encode them from complementary modality perspectives, thereby capturing both semantic and structural information. This is achieved through a parallel feature extraction pipeline with three concurrent streams: (1) Table-as-Text encoding: The raw table $\mathcal{T}$ is serialized into text and processed by the encoder $\mathcal{E}_t$ of the Table-as-Text model $\mathcal{M}_t$, yielding a feature vector $\mathbf{e}_t$. (2) Table-as-Image encoding: A screenshot of $\mathcal{T}$ is fed into the encoder $\mathcal{E}_v$ of the Table-as-Image model $\mathcal{M}_v$, producing $\mathbf{e}_v$. (3) Query encoding: The query $q$ is transformed into an embedding $\mathbf{e}_q$ using a text embedding model $\mathcal{E}_q$. We apply modality-specific pooling to convert encoder outputs into fixed-dimensional vectors,

which are concatenated to form a unified multimodal representation for the gating network:

$$
\begin{aligned}
\mathbf{e}_t &= \mathcal{E}_t(\text{Serialize}(\mathcal{T})) \\
\mathbf{e}_v &= \mathcal{E}_v(\text{Screenshot}(\mathcal{T})) \\
\mathbf{e}_q &= \mathcal{E}_q(q) \\
\mathbf{x} &= [\mathbf{e}_q, \mathbf{e}_t, \mathbf{e}_v]
\end{aligned}
\tag{1}
$$

Here, $\mathbf{x}$ serves as the multimodal joint representation by concatenating the three embeddings, which is subsequently passed into the gating network to determine the path of dynamic inference. Note that $\mathcal{E}_t$ and $\mathcal{E}_v$ only activate a small part of the corresponding expert models, constituting a minor computational overhead (i.e., 7.15% of the Table-as-Text model's and 7.63% of the Table-as-Image model's total parameters). Detailed analyses of the embedding construction and parameter breakdown are provided in Appendix A.3.1.

### 3.3 GATING NETWORK AND POLICY TRAINING

As discussed in the introduction and confirmed by our experiments, not every table–query pair benefits from fusing both modalities, as this can introduce redundant or even conflicting signals. To address this, TableDART employs a lightweight gating network that dynamically selects the inference path. This design both maximizes the effective use of information and reduces computational cost compared to always invoking all modality models.

Formally, the gating network $\mathcal{G}$ is implemented by a lightweight MLP with few trainable parameters (see Appendix A.3.2 for architectural details). It takes as input the multimodal representation $\mathbf{x}$ from Eq. 1 and outputs the raw logits $\mathbf{z}$ for each processing path (Text-only, Image-only, or Fusion):

$$
\mathbf{z} = \mathcal{G}(\mathbf{x}). \tag{2}
$$

To ensure that $\mathcal{G}$ selects the most appropriate strategy for each table-query pair, balancing both predictive accuracy and resource efficiency, we design the following training objective:

$$
\mathcal{L}_{\text{total}} = \mathcal{L}_{\text{task}} + \lambda \mathcal{L}_{\text{resource}}, \tag{3}
$$

where $\lambda$ controls the strength of the resource-aware regularization.

The task loss $\mathcal{L}_{\text{task}}$ encourages the gating network to prioritize paths with strong empirical performance. For each training instance, we pre-compute a vector of binary scores $\mathbf{s} = (s_1, s_2, s_3) \in \{0, 1\}^3$, where each component $s_k$ is 1 if the $k$-th inference path produces a correct answer, and 0 otherwise. This method naturally handles instances where multiple paths are correct, as each successful path receives a score of 1. To form the training signal, this score vector is converted into a soft target probability distribution using a temperature-controlled softmax. Similarly, the gating network's raw logits $\mathbf{z}$ are converted into a predicted distribution. The task loss then minimizes the KL divergence between these two distributions:

$$
\mathcal{L}_{\text{task}} = \text{KL}(\text{softmax}(\mathbf{s}/\tau) \,\|\, \text{softmax}(\mathbf{z}/\tau_g)), \tag{4}
$$

where $\tau_g, \tau$ are temperature parameters. We use the standard forward KL divergence, which encourages the gating network to assign probability to all empirically successful paths.

Beyond empirical performance on table understanding, we also account for inference overhead. To prevent over-reliance on costly strategies, we introduce a resource-aware regularizer:

$$
\mathcal{L}_{\text{resource}} = \text{softmax}(\mathbf{z}/\tau_g)^T \mathbf{c}, \tag{5}
$$

where $\mathbf{c}$ is an empirically measured cost vector for each inference path. By computing the expected cost of the routing policy, the resource loss $\mathcal{L}_{\text{resource}}$ penalizes high-cost inference routes, especially on simple tasks where $\mathcal{L}_{\text{task}}$ sees minimal improvements from over-complex paths. As such, it guides the gating policy toward more balanced choices, alleviating the information redundancy problem. The full training procedure, including the detailed protocol for measuring the cost vector $\mathbf{c}$, is provided in Appendix A.5.

## 3.4 Adaptive Inference

The adaptive inference process begins with a deterministic routing decision guided by the trained gating network. For each query-table pair, the network produces a vector of raw logit scores, and the framework selects the single, optimal path corresponding to the highest score for execution. Once the path is selected, inference proceeds accordingly. TableDART supports three inference paths: Text-only, Image-only, and Fusion. The first two directly reuse existing single-modality expert models: the Table-as-Text model $\mathcal{M}_t$ and the Table-as-Image model $\mathcal{M}_v$. If the gating network selects either, TableDART simply continues the inference process from the intermediate representation of the chosen model. For example, when the gate selects the Text-only path, TableDART forwards $\mathcal{M}_t$'s remaining layers using the intermediate encoding $\mathbf{e}_t$ to produce the final output.

For the Fusion path, we design a novel, training-free LLM-based Fusion agent that synthesizes the final answer by reasoning over the outputs of the two single-modality experts, $\mathcal{M}_t$ and $\mathcal{M}_v$. Specifically, when the gating network selects the Fusion path, both $\mathcal{M}_t$ and $\mathcal{M}_v$ are at first executed to generate results $r_t$ and $r_v$, along with their auxiliary outputs $a_t$ and $a_v$. These, together with the original table $\mathcal{T}$, are passed to the Fusion agent. The Fusion agent operates in two possible roles depending on its reasoning process: (1) **Arbitrator**: If $\mathcal{M}_t$ and $\mathcal{M}_v$ produce conflicting results, the agent resolves the disagreement, similar in spirit to aggregating a consensus from multiple sources (Hung et al., 2018), by selecting the more reliable answer according to its confidence. (2) **Rescuer**: If the agent believes both models provide uncertain or low-confidence outputs, it synthesizes a new, more accurate answer by jointly reasoning over their partial evidence. A detailed description of the Fusion agent's implementation and prompt design is provided in Appendix A.4.

## 4 Experiments

### 4.1 Experimental Setup

**Evaluation Datasets and Metrics.** We evaluate TableDART on seven diverse benchmarks across two tasks: Table Question Answering (TQA) and Table Fact Verification (TFV). For TQA, which requires interpreting complex queries (Ren et al., 2020; 2021) to perform grounded reasoning over tabular evidence, we use five benchmarks: WTQ (Pasupat & Liang, 2015), TABMWP (Lu et al., 2023), TAT-QA (Zhu et al., 2021), HiTab (Cheng et al., 2022), and FeTaQA (Nan et al., 2022). For TFV, we use TabFact (Chen et al., 2020) and InfoTabs (Gupta et al., 2020). Following the established protocol (Zheng et al., 2024; Liu et al., 2025), we use Accuracy for WTQ, TABMWP, TAT-QA, HiTab, TabFact, InfoTabs, while using BLEU score for FeTaQA as it is a generative free-form response task. Detailed statistics and further experimental setup details are provided in Appendix A.5.

**Baselines.** We compare TableDART against a comprehensive set of baselines, including (1) Table-as-Text models: Llama-2-7B (Touvron et al., 2023), Llama3-Instruct-8B (Dubey et al., 2024), TableLlama-7B (Zhang et al., 2024), and TableGPT2-7B (Su et al., 2024); (2) Table-as-Image models: MiniGPT-4-7B (Zhu et al., 2024a), mPLUG-Owl-7B (Ye et al., 2023), mPLUG-Owl2-7B (Ye et al., 2024), LLaVA v1.5-7B (Liu et al., 2024), Table-LLaVA-7B (Zheng et al., 2024), Qwen-VL-7B (Bai et al., 2023), InternLM-XComposer2-7B (Zhang et al., 2023), Monkey-7B (Li et al., 2024), TabPedia-7B (Zhao et al., 2024), SynTab-LLaVA-7B (Zhou et al., 2025), MiniCPM-V-2.6-8B (Yao et al., 2024), Qwen2.5-VL-7B (Bai et al., 2025), and Ovis2-8B (Lu et al., 2024); (3) Table-as-Multimodality models: HIPPO-8B (Liu et al., 2025) and Google Gemini 2.0 Flash (Comanici et al., 2025).

We employ TableGPT2-7B and Ovis2-8B as TableDART's primary single-modality models. To demonstrate generalization, we additionally evaluate a variant by replacing the Table-as-Image model Ovis2-8B with Qwen2.5-VL-7B while retaining TableGPT2-7B. We use Google Gemini 2.0 Flash to implement the Fusion agent. Comparing TableDART against these base models directly demonstrates the effectiveness of our framework. Appendix A.5.2 provides a detailed rationale for model selection and clarifies the sourcing of baseline results adopted from previous studies. Notably, TableDART is a general framework that can be seamlessly integrated with a wide range of existing LLMs, VLMs and MLLMs.

**Implementation Details.** Our training set is a 10,000-sample mixture constructed by sampling from five diverse table understanding benchmarks, following the protocol of prior work (Liu et al., 2025).

Table 1: Main performance comparison on seven benchmarks for Table Question Answering (TQA) and Table Fact Verification (TFV). The Average column shows the mean accuracy across all benchmarks that use Accuracy as the evaluation metric. The **best** and second-best results are marked.

| Method | TQA | | | | | TFV | | Summary |
|---|---|---|---|---|---|---|---|---|
| | WTQ (Acc.) | TABMWP (Acc.) | TAT-QA (Acc.) | HiTab (Acc.) | FeTaQA (BLEU) | TabFact (Acc.) | InfoTabs (Acc.) | Average (Acc.) |
| *Constituent Models of TableDART* | | | | | | | | |
| TableGPT2-7B *(Text-only Path)* | 61.42 | 83.87 | 50.39 | 70.27 | 28.97 | 77.80 | 71.07 | 69.14 |
| Ovis2-8B *(Image-only Path)* | 58.76 | 87.00 | 47.67 | 68.59 | 34.70 | 80.80 | 74.11 | 69.49 |
| *Table-as-Text Baselines* | | | | | | | | |
| Llama-2-7B | 16.39 | 22.82 | 13.73 | 10.72 | 10.93 | 9.20 | 38.92 | 18.63 |
| Llama3-Instruct-8B | 21.24 | 42.01 | 13.08 | 6.97 | 12.66 | 73.89 | 54.00 | 35.20 |
| TableLlama-7B | 24.97 | 10.10 | 19.04 | 46.57 | **38.38** | 79.37 | 46.57 | 37.77 |
| *Table-as-Image Baselines* | | | | | | | | |
| MiniGPT-4-7B | 0.90 | 0.22 | 0.13 | 0.20 | 0.39 | 0.00 | 0.10 | 0.26 |
| mPLUG-Owl-7B | 0.62 | 1.76 | 0.13 | 0.25 | 7.42 | 7.46 | 5.53 | 2.63 |
| mPLUG-Owl2-7B | 0.67 | 6.83 | 0.39 | 0.13 | 11.91 | 8.21 | 26.19 | 7.07 |
| LLaVA v1.5 | 1.24 | 6.05 | 2.97 | 2.03 | 8.24 | 18.90 | 28.31 | 9.92 |
| Table-LLaVA-7B | 18.43 | 57.78 | 12.82 | 10.09 | 25.60 | 59.85 | 65.26 | 37.37 |
| Qwen-VL-7B | 0.09 | 3.30 | 0.13 | 0.06 | 0.45 | 1.12 | 0.65 | 0.89 |
| InternLM-XComposer2-7B | 0.05 | 0.06 | 0.26 | 0.12 | 2.62 | 1.19 | 1.11 | 0.46 |
| Monkey-7B | 19.07 | 13.26 | 12.31 | 6.41 | 3.41 | 22.56 | 22.11 | 15.95 |
| TabPedia-7B | 23.53 | 10.66 | 13.08 | 6.54 | 14.31 | 35.49 | 2.43 | 15.29 |
| SynTab-LLaVA-7B | 39.59 | **88.30** | 51.94 | 35.66 | 35.45 | 70.78 | 69.42 | 59.28 |
| MiniCPM-V-2.6-8B | 47.97 | 83.68 | 51.55 | 56.53 | 32.68 | 78.48 | 73.03 | 65.21 |
| Qwen2.5-VL-7B | 54.37 | 63.69 | 51.94 | 62.69 | 10.99 | 75.81 | 70.13 | 63.11 |
| *Table-as-Multimodality (MLLM-based) Baselines* | | | | | | | | |
| HIPPO-8B | 55.77 | 87.50 | 60.75 | 63.00 | 33.18 | **82.27** | 75.74 | 70.84 |
| Gemini 2.0 Flash | 63.56 | 46.29 | 35.62 | 60.41 | 10.57 | 81.33 | 54.31 | 56.92 |
| *Table-as-Multimodality (Dynamic Adaptive Routing)* | | | | | | | | |
| TableDART (TableGPT2-7B + Qwen2.5-VL-7B) | 69.29 | 72.61 | 59.07 | 71.13 | 29.87 | 77.94 | 71.46 | 70.25 |
| **TableDART (TableGPT2-7B + Ovis2-8B)** | **70.58** | 84.54 | **62.05** | **74.37** | 36.11 | 81.37 | **76.22** | **74.86** |

We train only the lightweight MLP gating network while keeping all large LLM models frozen. The complete details regarding our data construction, hyperparameter settings, and computational environment are provided in Appendix A.5.

## 4.2 EFFECTIVENESS ANALYSIS

**State-of-the-Art Performance.** Table 1 shows the comparison results of TableDART with baselines. From the results, we can observe that: (1) Compared to all single-modality based methods, TableDART consistently achieves superior or highly competitive results, including surpassing its constituent models like TableGPT2-7B and Ovis2-8B, demonstrating that the framework successfully combines their strengths to achieve a result neither could reach alone. (2) Compared to multi-modality based methods, TableDART's dynamic routing mechanism proves more effective than the MLLM-based HIPPO, outperforming it on five of the seven benchmarks. (3) TableDART outperforms the powerful Fusion agent model (Google Gemini 2.0 Flash), confirming that the performance gains are driven by the intelligent routing mechanism, not just the capacity of the Fusion agent's backbone. (4) TableDART demonstrates strong **generalization** across backbone models. When instantiated by replacing the Table-as-Image model with Qwen2.5-VL-7B while retaining TableGPT2-7B, the framework achieves a substantial +7.14% average accuracy gain over the single-modality Qwen2.5-VL baseline, validating its model-agnostic effectiveness. Furthermore, the 'Average' column summarizes the average performance, showing that TableDART achieves the strongest results among all baselines, surpassing the best multi-modality tabular model, HIPPO-8B, by a decisive **+4.02%** in average accuracy and **+2.93** points in BLEU score.

**Zero-Shot Generalization Performance.** To assess TableDART's generalization capability, we compare it with the strongest MLLM-based baseline, HIPPO-8B, in a zero-shot setting on unseen datasets excluded from training. As shown in Table 2, TableDART demonstrates superior generalization. It achieves a **+18.05%** accuracy improvement and a +2.93 BLEU score gain over HIPPO-8B. Notably, TableDART maintains nearly identical performance on seen (74.95%) and unseen datasets (74.37%), whereas HIPPO-8B suffers a sharp drop from 72.41% to 63.00%. This substantial advantage highlights that TableDART learns a genuinely generalizable routing policy, capable of adapting to new tables rather than overfitting to the training distribution.

Table 2: Generalization performance on seen versus unseen datasets. Seen datasets were included in the training mixture, while unseen datasets were excluded from training.

| Method | Seen Datasets (Avg. Acc.) | Unseen Datasets (Avg. Acc. / BLEU) |
|---|---|---|
| HIPPO-8B *(Best MLLM-based Baseline)* | 72.41 | 63.00 / 33.18 |
| TableDART | 74.95 | 74.37 / 36.11 |
| **Improvement** | **+3.51%** | **+18.05% / +2.93 pts** |

## 4.3 EFFICIENCY ANALYSIS

**Training Efficiency.** TableDART demonstrates exceptional training efficiency rooted in its architectural design. Our framework freezes all large constituent models, requiring only the training of a lightweight MLP gating network with just **2.59M** parameters. This contrasts sharply with MLLM-based approaches like HIPPO, which fine-tune their 8B backbone model using LoRA (Hu et al., 2022). For their model architecture and LoRA configuration, the total number of trainable parameters reaches **25.87M**, meaning that HIPPO trains nearly **10 times more** parameters than our entire framework. This vast difference highlights the profound training efficiency and the plug-and-play scalability of our modular design.

**Inference Efficiency.** To quantify the practical benefits of dynamic adaptive routing, we benchmark TableDART's optimal configuration ($\lambda = 0.15$) against a Non-Adaptive Fusion baseline that processes every instance via the full multimodal pipeline. Both settings use identical backbone models and Fusion agents, ensuring a fair comparison to isolate the efficiency gains of our adaptive policy. The detailed evaluation protocol is in Appendix A.6. The results, presented in Table 3, demonstrate substantial efficiency gains. For example, TableDART achieves an average reduction of **24.5%** in latency compared to the Non-Adaptive Fusion baseline, decreasing the mean inference time from 2.92s to 2.20s per sample. These improvements are a direct consequence of TableDART's learned routing policy, which reserves the computationally expensive Fusion path for instances that truly require multimodal reasoning, while assigning simpler cases to the more efficient unimodal paths. As shown in Appendix A.7 and Appendix B.3, the model routes multimodally complex instances to the Fusion path while assigning simpler ones to the appropriate unimodal path. In particular, Figure 6 shows that 34.8% of TAT-QA instances cannot be solved by either single-modality model alone, and Figure 10 confirms that the learned policy routes 88.7% of these instances to the powerful Fusion path. In contrast, for simpler datasets such as TABMWP, where single-modality models are highly effective, the routing policy assigns 97.2% of the instances to the more efficient Image-only path. Overall, TableDART's inference efficiency derives from its intelligent dynamic routing mechanism, which preserves effectiveness while substantially reducing unnecessary computational cost.

Table 3: Inference efficiency for TableDART with its Dynamic Adaptive Routing policy versus the Non-Adaptive Fusion baseline. TPS is short for Tokens per Second. "↓" indicates lower is better, while "↑" indicates higher is better.

| Dataset | Dynamic Adaptive Routing | | Non-Adaptive Fusion | |
|---|---|---|---|---|
| | Latency (s) ↓ | TPS ↑ | Latency (s) ↓ | TPS ↑ |
| WTQ | 2.39 | 8.90 | 2.84 | 1.33 |
| TABMWP | 1.81 | 26.09 | 2.53 | 1.04 |
| TAT-QA | 2.35 | 22.36 | 3.17 | 2.22 |
| HiTab | 2.56 | 13.97 | 3.19 | 1.70 |
| FeTaQA | 1.93 | 18.98 | 2.99 | 11.11 |
| TabFact | 2.21 | 16.48 | 2.91 | 4.43 |
| InfoTabs | 2.17 | 17.60 | 2.78 | 5.09 |
| **Average** | **2.20** | **17.77** | 2.92 | 3.85 |

## 4.4 ANALYSIS OF EACH INFERENCE PATH

To better understand the contribution of each inference path in TableDART, we categorize instances based on whether the single-modality or Fusion paths could produce a correct solution. The statis-

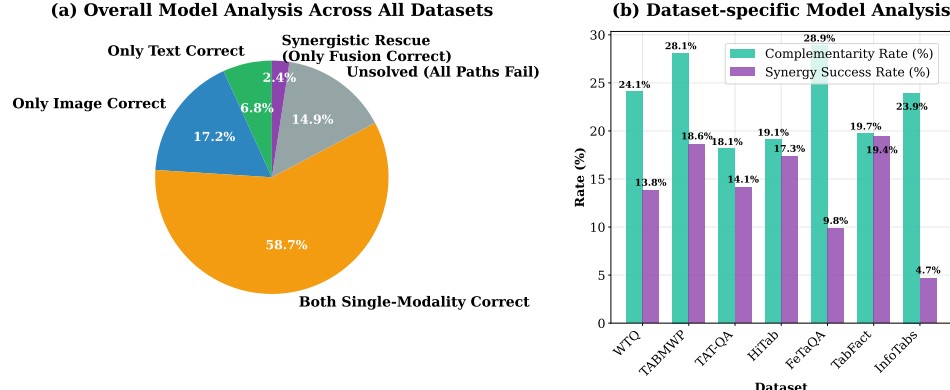

Figure 2: Performance analysis of inference paths. (a) The chart counts instances based on which path(s), if any, produced a correct answer, across all datasets. (b) A per-dataset analysis of two key metrics: the Complementarity Rate, which is the percentage of instances where correctness is achieved by only one of the two single-modality models, and the Synergy Success Rate, which measures the fraction of hard cases (instances where both single-modality models fail) that are successfully resolved by the Fusion path.

tics in Figure 2 provide three key insights. (1) Unimodal paths have complementary performance, as 24.0% of test instances fall into a complementarity zone where only single-modality paths generate correct answers (17.2% for the image-only path vs. 6.8% for the text-only path) as shown in Figure 2(a). This underscores the importance of maintaining distinct single-modality inference paths. Moreover, the complementarity rate varies across data sets according to Figure 2(b), highlighting the need for instance-specific dynamic routing. (2) Hard cases, where both single-modality inference paths fail, account for 17.3% of the test set. While 14.9% of these remain unsolved, the Fusion path successfully rescues an additional 2.4% (Figure 2a). This capability is further reflected in the Synergy Success Rate (i.e., the proportion of hard cases resolved by Fusion), which averages 14.0% across benchmarks (Figure 2b). These results confirm that our Fusion agent is not only an Arbitrator between modalities but also a Rescuer, capable of synergistic reasoning beyond the limits of individual models. (3) Easy cases where both unimodal inference paths succeed make up the majority of the data, accounting for 58.7% of the test set (Figure 2a). For this substantial portion, multimodal fusion is unnecessary, as either single-modality path suffices to produce the correct answer. This result reinforces our core motivation for dynamic adaptive routing, rather than defaulting to multimodal fusion in all cases.

## 4.5 Ablation Study on Key Components

**Impact of Dynamic Gating.** We ablate the routing strategy to validate our dynamic approach by comparing it with two non-adaptive baselines in Table 4. The results first confirm the effectiveness of our dynamic strategy, as TableDART significantly outperforms the Random Routing baseline. Compared with the Non-Adaptive Fusion baseline, which uses the same backbone models as TableDART but applies a brute-force strategy where every instance is routed through our LLM Agent-based Fusion path, our method not only achieves comparable results on most benchmarks but also delivers clear gains on datasets such as TABMWP and HiTab. This indicates that TableDART's effectiveness is not solely attributable to the strong Fusion agent, but arises from the dynamic routing mechanism itself. As shown in Figure 10, the learned policy routes most instances to the appropriate unimodal paths and uses Fusion only when needed. In many cases, a single modality already provides sufficient information, and forcing unnecessary multimodal processing introduces extra cost and potential noise, which is why TableDART surpasses the Non-Adaptive Fusion baseline on several datasets. Overall, these observations demonstrate the balance of resource-efficiency and performance-effectiveness of our dynamic routing design.

**Impact of Resource-Aware Objective** ($\lambda$)**.** We analyze the effect of the resource loss weight $\lambda$, highlighting its two key roles in shaping the final routing policy. (1) **Effective Routing Policy Control.** As shown in Figure 3, $\lambda$ directly regulates the framework's behavior. For instance, reducing its

| Method | TQA | | | | | TFV | |
|---|---|---|---|---|---|---|---|
| | WTQ | TABMWP | TAT-QA | HiTab | FeTaQA | TabFact | InfoTabs |
| TableDART w/ Random Routing | 65.40 | 75.50 | 58.94 | 70.49 | 30.87 | 79.50 | 69.57 |
| TableDART w/ Non-Adaptive Fusion | **70.97** | 81.47 | **63.34** | 73.35 | 34.82 | **81.56** | **76.83** |
| TableDART (Dynamic Routing) | 70.58 | **84.54** | 62.05 | **74.37** | **36.11** | 81.37 | 76.22 |

Table 4: Ablation on routing strategies.

| Benchmark | $\lambda = 1.0$ | $\lambda = 0.15$ | $\lambda = 0.1$ | $\lambda = 0.05$ | $\lambda = 0.00$ |
|---|---|---|---|---|---|
| *Table Question Answering (TQA)* | | | | | |
| WTQ | 64.94 | 70.58 | 67.96 | 71.80 | **71.96** |
| TABMWP | 83.86 | 84.54 | 84.79 | 85.08 | **85.47** |
| TAT-QA | 61.79 | 62.05 | 63.21 | 64.12 | **64.38** |
| HiTab | 73.79 | **74.37** | 72.14 | 74.30 | 73.22 |
| FeTaQA | 35.28 | **36.11** | 35.37 | 34.34 | 35.87 |
| *Table Fact Verification (TFV)* | | | | | |
| TabFact | 77.30 | 81.37 | 79.55 | **81.39** | 79.81 |
| InfoTabs | 70.13 | **76.22** | 70.44 | 73.59 | 74.19 |
| **Avg. Acc.** | 71.97 | 74.86 | 73.02 | **75.05** | 74.84 |

Table 5: Impact of the resource loss weight ($\lambda$) on TableDART's performance. The **best** and second-best results are highlighted.

value increases reliance on the computationally expensive Fusion path, causing more queries to be routed to the Fusion inference path.

(2) **Regularization for Improved Generalization.** Table 5 shows that our proposed resource-aware objective also serves as an effective regularizer. The highest performance, both in terms of average results and across four of the seven benchmarks, is not attained with a purely performance-driven policy ($\lambda = 0.00$), but rather at a non-zero $\lambda$. By penalizing computational cost, the objective discourages the gate from over-relying on the Fusion path, a tendency that risks overfitting to training artifacts. Instead, it promotes a more balanced routing policy that generalizes more effectively. In addition, varying $\lambda$ has a direct impact on inference efficiency. As illustrated in Appendix B.4 (Figure 11 and Table 11), efficiency generally improves as $\lambda$ increases, since a larger penalty discourages unnecessary use of the Fusion path. However, the trend is not strictly monotonic, because each $\lambda$ induces a distinct routing policy that balances accuracy and computational cost differently. Based on this observation, we select $\lambda = 0.15$ for the final model, which achieves the second-best average accuracy (74.86%, within 0.19 points of the best) while requiring 8.4% less inference latency, representing a well-regularized operating point on the performance–efficiency frontier. This configuration enables a truly adaptive policy, which tailors routing to the task type, directing 97.2% of queries to the Image-only path for the structure-heavy TABMWP, while shifting to 67.5% Text-only for the semantics-focused InfoTabs (see Appendix B.3 and Fig. 10 for a full breakdown).

## 4.6 CASE STUDY

To qualitatively illustrate the sophisticated reasoning of our Fusion path, Figure 4 presents two representative cases that highlight the LLM agent's primary roles. (1) The first case demonstrates the agent as an Arbitrator. Tasked with a probability calculation, the Table-as-Text model fails by incorrectly summing a subset of the data for its denominator, while the Table-as-Image model reasons correctly. The Fusion agent resolves this conflict by validating both reasoning paths against the source table and selecting the correct output. (2) The second case showcases the Fusion agent as a Rescuer. Here, Fusion agent reasons that both single-modality models fail to identify the two required occupations, with each providing only one correct and one hallucinated answer. The agent exhibits true synergy by synthesizing a new, fully correct answer, combining the valid fragments from both failed outputs while discarding the errors. This synthesis demonstrates the Fusion agent's ability to generate novel, correct answers from incomplete and conflicting information from two single-modality models.

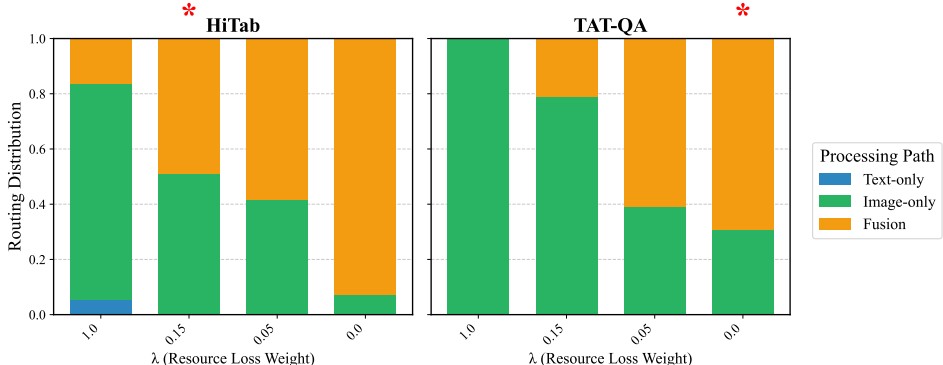

Figure 3: Inference path selection distribution vs. the resource loss weight ($\lambda$) on two representative benchmarks. Each bar shows the percentage of instances routed to the Text-only (blue), Image-only (green), and Fusion (orange) paths. A red star (*) marks the configuration with the highest performance for each dataset (see Table 5). This selection highlights TableDART's adaptability to diverse challenges. Full results on all seven datasets are provided in Appendix B.1.

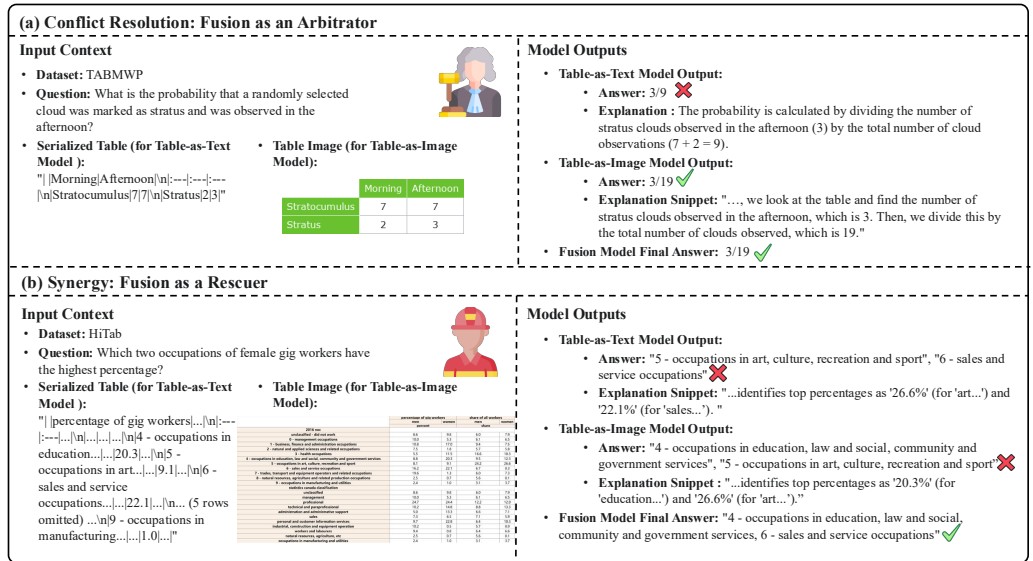

Figure 4: Case studies illustrate the key synthesis roles of the Fusion Model, which is implemented by an LLM agent. (a) As an **Arbitrator** (example from the TABMWP dataset), it resolves a conflict between a correct and an incorrect numerical reasoning path. (b) As a **Rescuer** (example from the HiTab dataset), it demonstrates synergy by synthesizing a correct answer from two distinct, incorrect outputs, showcasing its ability to combine partially correct reasoning fragments.

## 5 CONCLUSION

In this work, we introduced TableDART, a dynamic adaptive multi-modal routing framework for table understanding. We replace the conventional MLLM-based approach with a lightweight MLP-based gating network that learns to route each input to an optimal processing path: Text-only, Image-only, or Fusion. This instance-level routing is learned efficiently by training only the 2.59M-parameter gate while keeping the large base models entirely frozen, preserving their pre-trained capabilities and ensuring plug-and-play flexibility. Our experiments across seven diverse benchmarks demonstrate that TableDART consistently achieves strong performance, validating that its dynamic, resource-aware routing is a more effective and efficient paradigm for complex table reasoning tasks. Please refer to Appendix A.1 for a detailed reproducibility statement.

ACKNOWLEDGMENTS

The Australian Research Council partially supports this work under the streams of Future Fellowship (Grant No. FT210100624), the Discovery Project (Grant No. DP240101108 and DP260100326), and the Linkage Project (Grant No. LP230200892 and LP240200546).

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

# A APPENDIX

## A.1 REPRODUCIBILITY STATEMENT

We are committed to ensuring the reproducibility of our work. All source code, trained gating network checkpoints, and experiment configurations are provided at the following GitHub repository: https://github.com/xiaobo-xing/TableDART. Below is a summary of the key components for reproduction.

- **Model Architecture**: The overall architecture of TableDART is detailed in Section 3. Specific details of the embedding extraction pipeline and the MLP gating network are provided in Appendix A.3. The **primary** constituent models are TableGPT2-7B (Su et al., 2024) and Ovis2-8B (Lu et al., 2024), with Qwen2.5-VL-7B (Bai et al., 2025) additionally employed to validate generalization. Google's Gemini 2.0 Flash model (Comanici et al., 2025) implements the Fusion agent.

- **Training Procedure**: Our parameter-efficient training methodology for the gating network, including the resource-aware objective function, is described in Section 3.3. Full implementation details are provided in Appendix A.5.

- **Datasets and Preprocessing**: The datasets used for training and evaluation are described in Section 4. Our training mixture was constructed from five datasets: WTQ (Pasupat & Liang, 2015), TABMWP (Lu et al., 2023), TAT-QA (Zhu et al., 2021), TabFact (Chen et al., 2020), and InfoTabs (Gupta et al., 2020). We evaluate on these five datasets plus two zero-shot datasets: HiTab (Cheng et al., 2022) and FeTaQA (Nan et al., 2022). Our specific data construction protocol is detailed in Section 4.1 and Appendix A.5.

- **Hyperparameters and Computational Infrastructure**: A comprehensive list of all hyperparameters is provided in Table 10. All experiments were conducted on a single NVIDIA H100 80GB GPU. These computational resources were provided by the Bunya supercomputer (The University of Queensland Research Computing Centre, 2024), operated by The University of Queensland Research Computing Centre (RCC). Full details on the training pipeline, hyperparameters, and computational setup are located in Appendix A.5.

## A.2 LARGE LANGUAGE MODELS USAGE

Large Language Models were employed to fix the grammar issues in this paper. The authors remain fully responsible for all content.

## A.3 MODEL ARCHITECTURE DETAILS

### A.3.1 EMBEDDING ARCHITECTURE AND COMPUTATIONAL ANALYSIS

The gating network requires fixed-dimensional feature representations to enable consistent routing decisions across variable-length inputs. We employ different pooling strategies optimized for each modality's characteristics:

- **Table-as-Text Model Features**: 3,584 dimensions obtained through attention-masked mean pooling of TableGPT2-7B input embeddings. This approach weights each token by its attention importance, preventing padding tokens from diluting the semantic representation while preserving global table structure information essential for routing decisions.

- **Table-as-Image Model Features**: 6,144 dimensions derived from spatial-temporal pooling of Ovis2-8B visual tokenizer outputs. Mean pooling across spatial dimensions captures holistic visual layout characteristics rather than fine-grained spatial details, providing sufficient information for modality appropriateness assessment without overwhelming the routing mechanism with local visual features.

- **Question Embedding**: 384 dimensions from all-MiniLM-L6-v2 (Reimers & Gurevych, 2019) sentence transformer. This pre-trained model provides question-type agnostic semantic representations that complement the features from the single-modality models, enabling the gating network to learn routing patterns based on question semantics rather than model-specific encodings.

**Total Concatenated Dimension**: $3{,}584 + 6{,}144 + 384 = 10{,}112$ dimensions. This unified multimodal representation serves as the input to the MLP gating network, which learns to map these heterogeneous features to processing path selection probabilities.

**Parameter Analysis**: The embedding extraction phase utilizes only a small fraction of each model's total parameters: (1) The Table-as-Text model requires 545M parameters from the input embedding layer (152K vocab × 3,584 dimensions), representing 7.15% of TableGPT2-7B's 7.62B parameters; (2) The Table-as-Image model requires 682M parameters from the Vision Transformer component aimv2-huge-patch14-448 for visual feature extraction, representing 7.63% of Ovis2-8B's 8.94B parameters. This computational efficiency enables lightweight routing decisions while preserving rich semantic representations from the pre-trained models.

A.3.2 GATING NETWORK ARCHITECTURE SPECIFICATION

Following standard practice in dynamic routing and mixture-of-experts systems in the prior works (Lepikhin et al., 2021; Zhu et al., 2022; Xue & Marculescu, 2023), the MLP gating network implements a standard two-layer architecture designed for efficient and effective routing. The detailed forward pass is defined as:

$$\mathbf{h} = \text{ReLU}(\mathbf{W}_1\mathbf{x} + \mathbf{b}_1) \tag{6}$$

$$\mathbf{h}' = \text{Dropout}(\mathbf{h}, p = 0.1) \tag{7}$$

$$\mathbf{z} = \mathbf{W}_2\mathbf{h}' + \mathbf{b}_2 \tag{8}$$

where $\mathbf{x} \in \mathbb{R}^{10112}$ is the concatenated multi-modal input features, and $\mathbf{z} \in \mathbb{R}^3$ represents the final expert selection logits. During inference, these logits are converted to probabilities via a Softmax function to select the optimal expert.

**Implementation Details.** The hidden dimension is set to 256, a choice empirically found to balance representational capacity for learning complex routing patterns with computational efficiency. Dropout with a probability of $p = 0.1$ is applied after the hidden layer to prevent overfitting. We use the ReLU activation function for its computational efficiency and stable gradient properties. All weight matrices are initialized using Xavier uniform initialization (Glorot & Bengio, 2010), and bias vectors are initialized to zero to ensure stable training dynamics.

**Parameter Count.** The lightweight nature of our gating network is evident in its parameter count, detailed in Table 6. The total of **2.59M** trainable parameters is negligible compared to the billions of parameters in the frozen LLM/MLLM models, underscoring the high parameter efficiency of our TableDART framework.

| Component | Computation | Parameters |
|---|---|---|
| **First Layer** | | |
| Weight matrix $\mathbf{W}_1$ | $10,112 \times 256$ | 2,588,672 |
| Bias vector $\mathbf{b}_1$ | 256 | 256 |
| *First layer subtotal* | | 2,588,928 |
| **Second Layer** | | |
| Weight matrix $\mathbf{W}_2$ | $256 \times 3$ | 768 |
| Bias vector $\mathbf{b}_2$ | 3 | 3 |
| *Second layer subtotal* | | 771 |
| **Total Network Parameters** | | **2,589,699 ≈ 2.59M** |

Table 6: Detailed parameter breakdown of the MLP gating network.

A.4 FUSION PATH DETAILED DESIGN

This section provides the detailed design of the Fusion path, which serves as the sophisticated synthesis mechanism within the TableDART framework. As described in the main text, this component is implemented as an LLM agent that interfaces with Google's Gemini 2.0 Flash model via REST API calls. For each instance, the LLM agent receives four key inputs: (1) the original question, (2) the complete table data in a structured markdown format, (3) the output from the Table-as-Text model containing its answer and explanation, and (4) the output from the Table-as-Image model containing its answer and explanation. These inputs are formatted into a prompt whose logical structure is illustrated in Figure 5. To ensure output format compatibility across benchmarks, the prompt is dynamically adapted based on the target dataset. As detailed in Table 7, specific instructions are appended to the prompt to meet the unique formatting requirements of each benchmark, such as requiring JSON outputs for TabFact or full-sentence responses for FeTaQA.

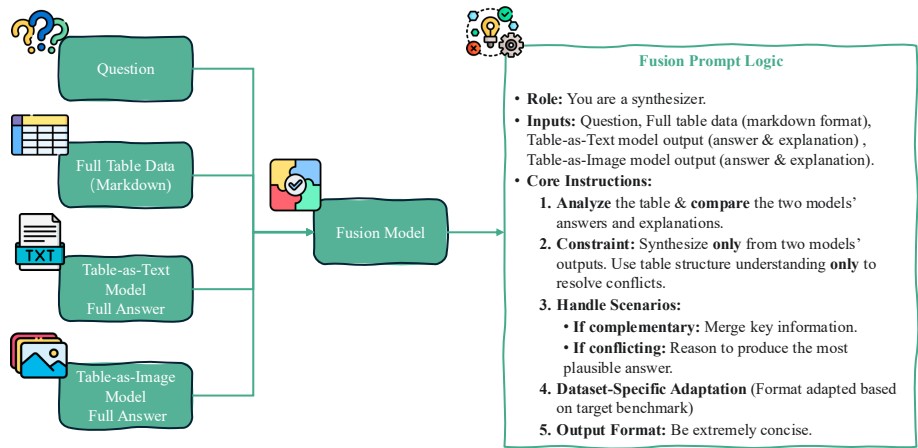

Figure 5: The data flow and logical structure of the prompt for the Fusion path's LLM agent. This structure guides the synthesis of a final answer from the outputs of the single-modality models.

| Dataset(s) | Additional Instruction |
|---|---|
| TabFact | Generate JSON response with "answer" field containing ["True"] or ["False"] |
| InfoTabs | Generate JSON response with "answer" field: ["Entail"], ["Contradict"], or ["Neutral"] |
| TabMWP | Output numeric answers without units when applicable |
| FeTaQA | Provide complete sentence responses, not keywords or phrases |
| WTQ, HiTab, TAT-QA | Standard JSON format with concise, direct answers |

Table 7: Dataset-specific prompt adaptations for the Fusion path's LLM agent.

## A.5 EXPERIMENTAL SETUP AND IMPLEMENTATION DETAILS

### A.5.1 DATASETS AND METRICS

Our experimental evaluation is conducted on seven diverse benchmarks across two primary tasks: Table Question Answering (TQA) and Table Fact Verification (TFV). We follow the established protocol of previous works (Zheng et al., 2024; Liu et al., 2025) for evaluation metrics, using Accuracy for most tasks and BLEU score for the generative FeTaQA benchmark. Table 8 provides a detailed breakdown of these datasets, including the number of instances used for our training mixture and the full size of the official test sets.

| Task Type | Dataset | Train | Validation | Test |
|---|---|---|---|---|
| Table Question Answering (TQA) | TABMWP | 2,000 | 300 | 7,686 |
| | WTQ | 2,000 | 300 | 4,344 |
| | HiTab | — | — | 1,586 |
| | TAT-QA | 2,000 | 300 | 772 |
| | FeTaQA | — | — | 2,003 |
| Table Fact Verification (TFV) | TabFact | 2,000 | 300 | 6,845 |
| | InfoTabs | 2,000 | 300 | 5,400 |
| **Total** | | **10,000** | **1,500** | **28,636** |

Table 8: Statistics of the datasets used for our experiments. The 'Test' column reflects the official benchmark size, while the 'Train' and 'Validation' columns indicate the number of instances we sampled from the official training data to construct our training mixture, as detailed in Section 4.1.

### A.5.2 Comparative Models and Rationale

We compare TableDART against a comprehensive set of baselines, strategically selected to validate our core contributions across several axes:

- **Constituent Models:** To demonstrate that TableDART's performance stems from its dynamic framework, we evaluate its core constituent models as standalone baselines. These include: (1) the Table-as-Text model, TableGPT2-7B (Su et al., 2024); (2) the primary Table-as-Image model, Ovis2-8B (Lu et al., 2024); (3) the additional visual expert used for generalization, Qwen2.5-VL-7B (Bai et al., 2025); and (4) the backbone of our Fusion agent, Google Gemini 2.0 Flash (Comanici et al., 2025). We selected these backbone models because they are recent and strong representatives of their respective paradigms. Evaluating the single-modality experts in isolation establishes a performance baseline. Specifically, including Qwen2.5-VL-7B enables us to test the framework's generalization capabilities across different visual backbones, while comparing against Gemini as a standalone MLLM verifies that our performance lift arises from the routing mechanism rather than only the capacity of the Fusion agent alone.

- **MLLM-based Baselines:** To highlight the superiority of our dynamic adaptive routing paradigm over static, one-size-fits-all approaches, we compare against two key models. First, HIPPO (Liu et al., 2025), a recent model representing the MLLM paradigm that jointly processes both text and image representations for all inputs. Second, Google Gemini 2.0 Flash (Comanici et al., 2025). Since Gemini serves as the backbone for the LLM agent in our Fusion path, including it as a standalone MLLM baseline allows us to demonstrate that the observed improvements stem from our dynamic routing framework itself, rather than solely from the capacity of the Fusion agent's underlying model.

- **Broader Competitive Landscape:** To position TableDART within the broader field, we benchmark it against an extensive suite of single-modality baselines in the 7-8B parameter range. Our selection of Table-as-Text models includes generalist LLMs like Llama-2-7B (Touvron et al., 2023) and Llama3-Instruct-8B (Dubey et al., 2024), as well as the specialized TableLlama-7B (Zhang et al., 2024). The set of Table-as-Image MLLMs is equally comprehensive, featuring MiniGPT-4-7B (Zhu et al., 2024a), mPLUG-Owl-7B (Ye et al., 2023), mPLUG-Owl2-7B (Ye et al., 2024), LLaVA v1.5-7B (Liu et al., 2024), Table-LLaVA-7B (Zheng et al., 2024), Qwen-VL-7B (Bai et al., 2023), InternLM-XComposer2-7B (Zhang et al., 2023), Monkey-7B (Li et al., 2024), TabPedia-7B (Zhao et al., 2024), SynTab-LLaVA-7B (Zhou et al., 2025), Qwen2.5-VL-7B (Bai et al., 2025) and MiniCPM-V-2.6-8B (Yao et al., 2024).

For a comprehensive and fair comparison, we report several baseline results directly from their original publications. Specifically, the results for the following models are adopted from the HIPPO paper (Liu et al., 2025): Llama-2-7B, Llama3-Instruct-8B, TableLlama-7B, MiniGPT-4-7B, mPLUG-Owl-7B, mPLUG-Owl2-7B, LLaVA v1.5, Table-LLaVA-7B, MiniCPM-V-2.6-8B, Qwen-VL-7B, InternLM-XComposer2-7B, Monkey-7B, and the HIPPO model itself. Furthermore, the results for TableGPT2-7B are sourced from its proposal paper (Su et al., 2024), and the results for SynTab-LLaVA are sourced from Zhou et al. (2025). We also adopt the results for TabPedia and Ovis2-8B (except TABMWP dataset) from Jiang et al. (2025). All other model results not mentioned above were generated by our own experimental runs.

### A.5.3 Training and Implementation Details

**Training Data Construction.** Our training set is a 10,000-sample mixture, constructed by randomly sampling 2,000 instances from five datasets: TABMWP, WTQ, TAT-QA, TabFact, and InfoTabs (Lu et al., 2023; Pasupat & Liang, 2015; Zhu et al., 2021; Chen et al., 2020; Gupta et al., 2020). Following the protocol of HIPPO (Liu et al., 2025), FeTaQA and HiTab are excluded from training due to their non-accuracy-based evaluation metric (BLEU) and challenges in serializing complex hierarchical structures, respectively (Nan et al., 2022; Cheng et al., 2022). An additional 15% validation split (1,500 samples) is generated using the same random sampling procedure with a fixed seed to ensure reproducibility.

**Processing Path Cost Measurement Protocol.** The resource cost vector **c** used in our resource-aware objective is derived from empirical measurements to provide a realistic estimate of each path's computational expense. We measured these costs on a representative testbed of 70 samples (10 from each benchmark), averaging over 10 timed runs following 5 warm-up iterations. Our final cost metric offers a holistic balance of latency (seconds per instance) and throughput (tokens per second), defined as: Cost $= 0.5 \times$ (Avg Latency) $+ 0.5 \times (1.0/\text{Avg TPS})$. For the Fusion path, its latency assumes parallel execution of its base models plus a 0.3s API average overhead, reflecting a realistic deployment scenario. The resulting empirically derived cost values are summarized in Table 9.

Table 9: Breakdown of the processing path cost calculation. The Fusion path's latency assumes parallel execution of the two single-modality models (taking the max latency) plus a measured API overhead. Its TPS is consequently bottlenecked by the slower Table-as-Image model.

| Processing Path | Avg. Latency (s) | Avg. TPS | Final Cost (c) |
|---|---|---|---|
| Text-only | 1.445 | 44.19 | 0.73 |
| Image-only | 1.559 | 18.78 | 0.81 |
| Fusion | 1.859 | 18.78 | 0.96 |

**Hyperparameter Configuration.** Table 10 summarizes the complete hyperparameter settings used for training the gating network. The resource loss weight $\lambda = 0.15$ is selected based on extensive ablation studies (Section 4.5), which demonstrate optimal performance-efficiency trade-offs at this value. These hyperparameters are selected through preliminary experiments to balance training stability and convergence speed. The target temperature $\tau = 0.3$ creates sufficiently sharp target distributions while preventing degenerate solutions, while the gate temperature $\tau_g = 1.0$ maintains standard softmax behavior during training.

| Parameter | Value |
|---|---|
| Learning Rate | 1e-4 |
| Batch Size | 8 |
| Gradient Accumulation Steps | 4 |
| Effective Batch Size | 32 |
| Weight Decay | 0.01 |
| Gradient Clipping (Max Norm) | 1.0 |
| Hidden Dimension | 256 |
| Dropout Rate | 0.1 |
| Target Temperature ($\tau$) | 0.3 |
| Gate Temperature ($\tau_g$) | 1.0 |
| Resource Loss Weight ($\lambda$) | 0.15 |
| LR Warmup Ratio | 0.05 |
| Training Epochs | 1 |

Table 10: Complete hyperparameter configuration for TableDART training.

**Training Pipeline.** Expert model parameters remain frozen throughout training, with only the lightweight 2-layer MLP gating network being optimized. This parameter-efficient approach dramatically reduces computational requirements compared to joint fine-tuning alternatives. The learning rate follows a cosine annealing schedule with 5% linear warmup, starting from zero and reaching the maximum learning rate of 1e-4 after the warmup period. We employ the AdamW optimizer with weight decay 0.01 to prevent overfitting of the gating network. Gradient accumulation over 4 steps achieves an effective batch size of 32 while maintaining memory efficiency on single-GPU training. Gradient clipping with maximum norm 1.0 ensures training stability, particularly important given the dynamic target generation process.

**Computational Configuration and Convergence.** Training is conducted on a single NVIDIA H100 80GB GPU with mixed precision optimization. Expert models utilize bfloat16 precision to reduce memory consumption, while the gating network maintains float32 precision for numerical stability during gradient computation. The complete training process requires approximately 13.5

hours to complete one epoch. Training converges rapidly within this single epoch, which we empirically determined to be optimal to prevent overfitting. The model is evaluated on the held-out validation set, and the checkpoint with the highest validation accuracy is saved as the best model for inference.

### A.6 EFFICIENCY BENCHMARK PROTOCOL

This section provides a detailed account of the protocol used for the efficiency analysis presented in Section 4.2. We outline the methodology for data sampling, performance measurement under a parallel assumption, and the precise definitions for all reported metrics to ensure full reproducibility.

#### A.6.1 BENCHMARK SETUP AND DATA SAMPLING

To create a representative and manageable testbed, we constructed an evaluation set via stratified random sampling from the seven benchmark test sets. We randomly sampled a balanced set of 50 instances from each dataset, resulting in a comprehensive benchmark suite of 350 unique samples. To ensure statistical stability, the entire measurement process was repeated three times with different random seeds, and all reported metrics are the average across these independent runs. All benchmarks were executed on a single NVIDIA H100 80GB GPU.

#### A.6.2 MEASUREMENT PROTOCOL AND PARALLEL ASSUMPTION

Our measurement protocol is designed to simulate a realistic parallel-processing deployment scenario. For each sample, the total inference latency is calculated by summing the durations of three sequential phases, with parallelism applied within phases where appropriate.

- **Phase 1: Parallel Feature Extraction.** The framework concurrently extracts embeddings from the input question, the serialized text table, and the table image. The duration of this phase is determined by the maximum latency among these three parallel operations: $T_{\text{phase1}} = \max(T_{\text{text\_embed}}, T_{\text{vision\_embed}}, T_{\text{question\_embed}})$.
- **Phase 2: Gating and Routing.** The concatenated embeddings are fed into the lightweight gating network to yield a routing decision. This phase is only applicable to TableDART; for the Non-Adaptive Fusion baseline, its duration is zero.
- **Phase 3: Generation and Fusion.** The execution path depends on the routing decision:
  - **For TableDART (Unimodal Path):** If the Text-only or Image-only path is selected, the duration is simply the generation time of the chosen model ($T_{\text{text\_gen}}$ or $T_{\text{vision\_gen}}$).
  - **For TableDART (Fusion Path) and the Non-Adaptive Fusion baseline:** Both the Table-as-Text and Table-as-Image models perform generation in parallel, followed by a call to the Fusion API. The duration is the maximum of the two generation times plus the API call latency: $T_{\text{phase3}} = \max(T_{\text{text\_gen}}, T_{\text{image\_gen}}) + T_{\text{fusion\_api}}$.

The total latency for each sample is the sum of these three phases: $L_{\text{parallel}} = T_{\text{phase1}} + T_{\text{phase2}} + T_{\text{phase3}}$.

#### A.6.3 METRIC DEFINITIONS

We use the following two primary metrics to report efficiency:

- **Latency (s):** The total time in seconds to process a single sample, calculated as $L_{\text{parallel}}$ under the parallel assumption described above. This is our primary metric for comparing the end-to-end speed of different frameworks. Lower values are better.
- **Tokens per Second (TPS):** A measure of throughput, calculated by dividing the number of generated output tokens by the total parallel latency. The token count is estimated using the Text Expert's tokenizer. Higher values are better.

### A.7 DETAILED ARCHITECTURAL ANALYSIS

This section provides detailed visualizations to supplement the summary analysis in Section 4.4. Figure 6 offers a full, per-dataset breakdown of the performance overlap between the single-modality

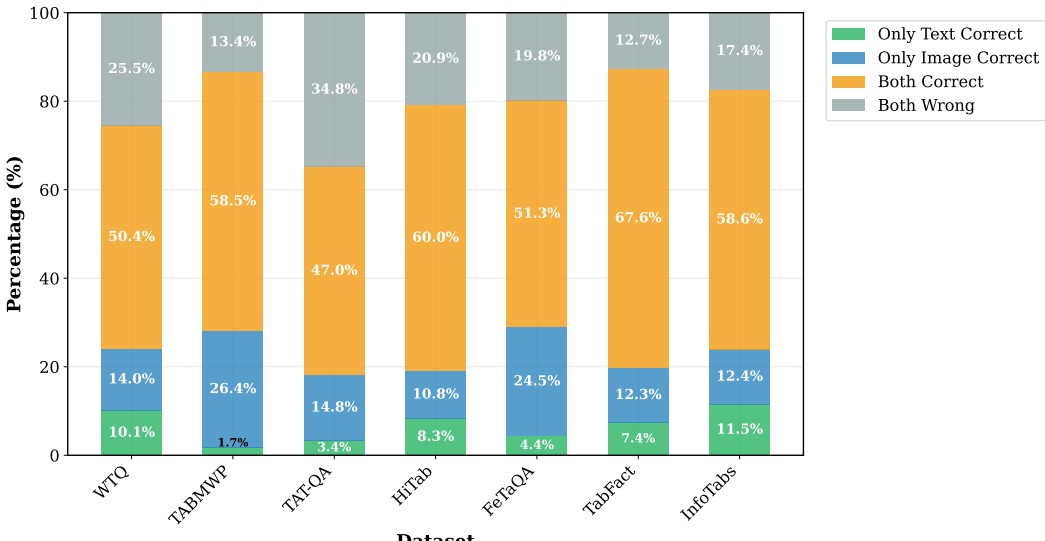

Figure 6: Per-dataset breakdown of performance overlap between the single-modality models. The stacked bars show the percentage of instances solved by only the Table-as-Text model, only the Table-as-Image model, both, or neither. The distribution varies significantly across benchmarks, highlighting the need for adaptive routing.

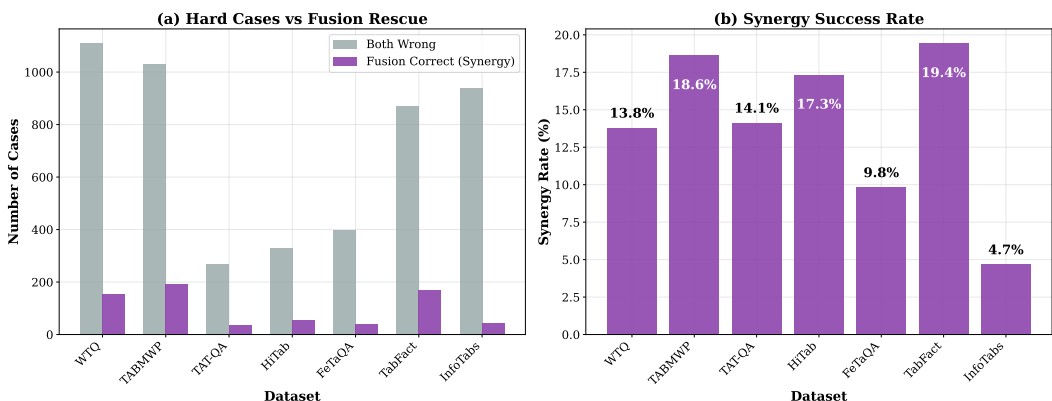

Figure 7: Detailed analysis of Fusion Path Synergy. (a) The absolute number of hard cases (where both base models failed) and the number of those cases successfully rescued by the Fusion path (synergy). (b) The Synergy Success Rate (Fusion Path Correct / Both Base Models Wrong) for each dataset demonstrates the consistent effectiveness of the Fusion path.

models. Figure 7 provides a more detailed view of the synergy analysis, showing both the absolute number of hard cases and the corresponding synergy success rates for each benchmark. These detailed charts serve as the underlying evidence for the aggregate statistics and trends discussed in the main paper.

## B   FURTHER IN-DEPTH ANALYSIS

### B.1   FULL GATING DECISION ANALYSIS ON ALL DATASETS

To supplement the analysis in the main paper, this section provides the complete visualizations of the processing path selection distribution for all seven benchmarks. Figure 3 in the main text highlights two representative datasets, while Figure 8 below shows the results on the remaining five.

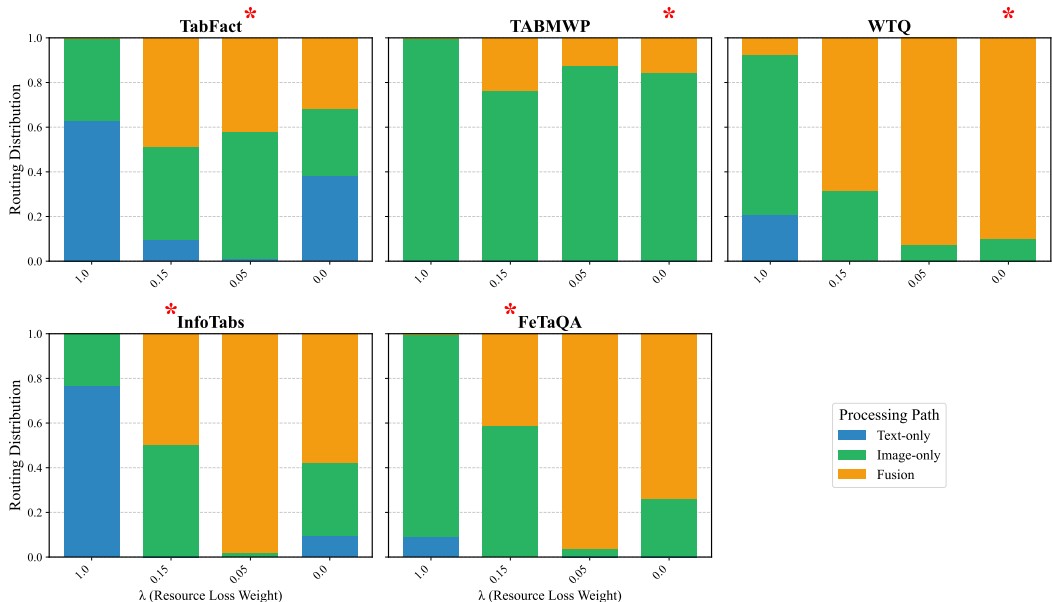

Figure 8: Processing path selection distribution vs. the resource loss weight ($\lambda$) on the five additional benchmarks. A red star (*) marks the best-performing configuration for each dataset. The data demonstrates consistent adaptive behavior across these diverse tasks.

The trends observed here are consistent with our main findings. For instance, on the math-heavy **TABMWP** dataset, the policy heavily favors the Image-only path. On the visually complex **HiTab** benchmark, the routing policy maintains a significant reliance on both the Image-only and Fusion paths across all $\lambda$ values. Finally, the generative **FeTaQA** task shows a strong preference for the Fusion path when performance is prioritized. These varied, dataset-specific behaviors further validate that TableDART successfully learns to adapt its strategy to the underlying characteristics of the data.

## B.2 ANALYSIS OF THE LEARNED ROUTING POLICY

Our final in-depth analysis dissects the sophisticated nature of the learned routing policy, revealing a crucial trade-off between aligning with a simple heuristic and achieving globally-optimal task performance. To investigate this, we adopt a behavioral analysis approach, a common technique in policy analysis and explainable AI (Guidotti et al., 2019), by defining a Heuristic Alignment Score. This metric measures the percentage of times TableDART's routing policy matches a simple, greedy heuristic: always choose the most cost-effective processing path (first Text-only, then Image-only, then Fusion) that is known to solve a given problem correctly. The procedure for calculating this metric is detailed in Algorithm 1. We then plot this alignment score against the final average task performance for each of our $\lambda$ configurations.

The result, shown in Figure 9, illustrates a clear non-linear relationship, resembling a Pareto frontier. We observe that a policy laser-focused on maximizing heuristic alignment (e.g., at $\lambda = 1.0$, the rightmost point) adheres most closely to the simple greedy rule but yields a suboptimal overall task performance. This demonstrates that a greedy, locally-focused strategy can be detrimental to the global objective.

Crucially, our chosen configuration, $\lambda = 0.15$, resides at the apex of this performance curve. Its modest Heuristic Alignment Score indicates that it has learned a more sophisticated, non-greedy policy. It quantitatively proves that the globally optimal strategy, learned via end-to-end training, involves strategically investing in more computationally expensive processing paths, even when a cheaper option is technically viable. This finding validates that TableDART learns a truly effective, globally-optimized routing policy that transcends simple heuristics.

---

**Algorithm 1** Heuristic Alignment Score Calculation

---

1: **Input:** Set of test samples $S$, Table-as-Text model results $R_{Text}$, Table-as-Image model results $R_{Image}$, TableDART's routing decisions $D_{TableDART}$.
2: **Output:** Heuristic Alignment Score $S_{align}$.
3:
4: $N_{aligned\_routes} \leftarrow 0$
5: $N_{total} \leftarrow |S|$
6:
7: **for** each sample $s_i \in S$ **do**
8:     $is\_text\_correct \leftarrow R_{Text}(s_i)$ is correct
9:     $is\_image\_correct \leftarrow R_{Image}(s_i)$ is correct
10:     $TableDART\_choice \leftarrow D_{TableDART}(s_i)$
11:
12:                                                                 ▷ Define the simple, greedy heuristic action
13:     **if** $is\_text\_correct$ **then**
14:         $heuristic\_choice \leftarrow$ "Text-only"
15:     **else if** $is\_image\_correct$ **then**
16:         $heuristic\_choice \leftarrow$ "Image-only"
17:     **else**
18:         $heuristic\_choice \leftarrow$ "Fusion"
19:     **end if**
20:
21:     **if** $TableDART\_choice = heuristic\_choice$ **then**
22:         $N_{aligned\_routes} \leftarrow N_{aligned\_routes} + 1$
23:     **end if**
24: **end for**
25:
26: $S_{align} \leftarrow (N_{aligned\_routes}/N_{total}) \times 100$
27: **return** $S_{align}$

---

### B.3 ANALYSIS OF PER-DATASET ROUTING POLICIES

To provide quantitative evidence of TableDART's adaptive capabilities, we analyze the routing decisions of our final model (trained with $\lambda = 0.15$) on each of the seven test benchmarks. Figure 10 visualizes the learned policy, revealing how the framework dynamically allocates resources based on the specific demands of each dataset.

The analysis highlights several key adaptive behaviors:

- **Adaptation to Modality Strengths:** The model demonstrates a profound understanding of modality-task alignment. For **TABMWP**, a benchmark requiring structural and mathematical reasoning, the policy routes an overwhelming **97.2%** of instances to the Image-only path. Conversely, for **InfoTabs**, a fact-verification task dependent on fine-grained semantics, the strategy shifts dramatically to favor the Text-only path (**67.5%**).

- **Adaptation to Task Difficulty:** On challenging benchmarks where single-modality models are likely to fail, such as **TAT-QA**, the policy learns to prioritize correctness by invoking the powerful but costly Fusion path for **88.7%** of cases. This demonstrates an ability to gauge task difficulty and escalate to a more robust strategy when necessary.

- **Balanced, Nuanced Strategies:** For benchmarks with a mix of challenges like **WTQ** and **FeTaQA**, the model learns a more balanced policy, distributing queries across all three paths. This indicates that the routing is not coarse-grained but makes fine-tuned, instance-level decisions.

In summary, this analysis provides compelling evidence that TableDART operates as a truly adaptive framework, which is the key to its state-of-the-art performance and efficiency.

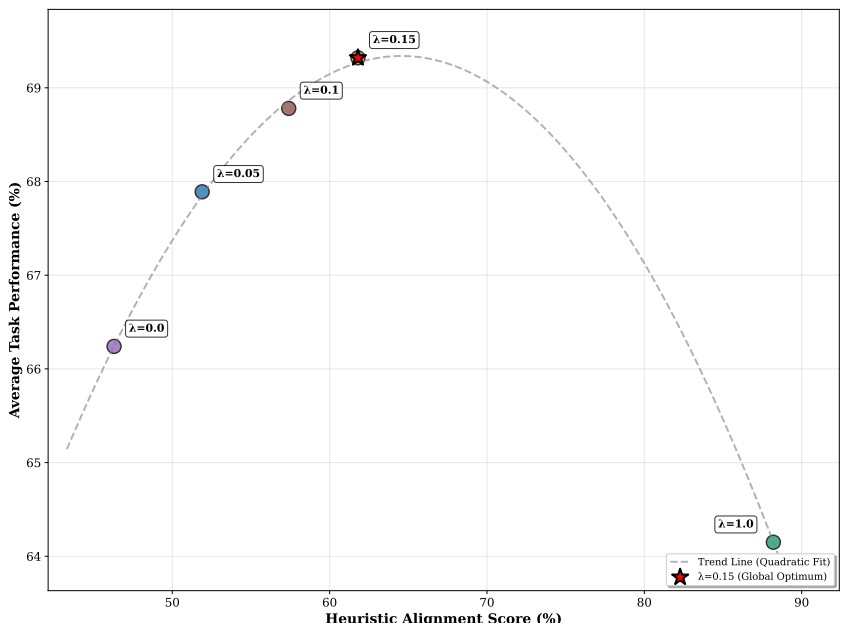

Figure 9: The trade-off between aligning with a simple greedy heuristic (x-axis) and achieving globally-optimal task performance (y-axis). The curve shows that strictly following the greedy heuristic can hurt final performance, and TableDART's best configuration ($\lambda = 0.15$) learns a superior, non-greedy policy.

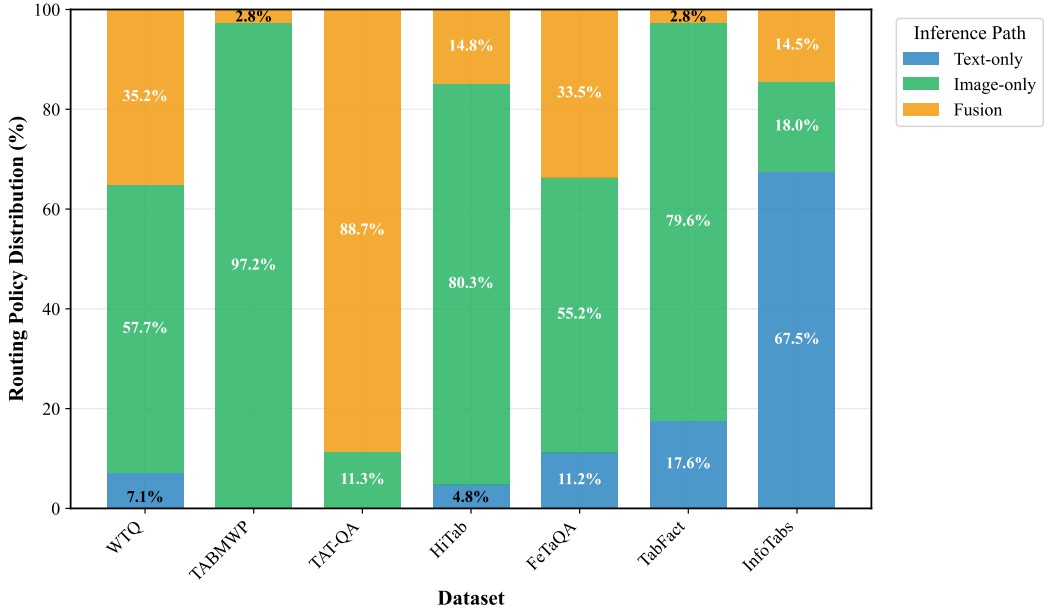

Figure 10: The learned routing policy distribution for each benchmark using the final TableDART model ($\lambda = 0.15$). The distinct strategies across datasets provide strong evidence of the framework's adaptive nature.

## B.4 EFFECT OF $\lambda$ ON INFERENCE EFFICIENCY

To more clearly characterize how the resource-aware objective influences efficiency, we report empirical latency and throughput measurements across different values of $\lambda$ in Table 11. As shown, $\lambda$

has a direct impact on inference cost. While larger $\lambda$ generally encourages more efficient routing by discouraging unnecessary use of the Fusion path, the trend is not strictly monotonic. This is because each $\lambda$ induces a distinct routing policy that balances accuracy and computational cost differently.

Figure 11 visualizes this relationship by plotting average latency against average accuracy. The resulting curve illustrates a natural performance–efficiency frontier: smaller $\lambda$ values (e.g., 0.0 and 0.05) tend to overuse the Fusion path, while very large values (e.g., 1.0) favor efficiency at the cost of reduced accuracy. The configuration $\lambda = 0.15$ provides the best overall balance, achieving the second-highest average accuracy (within 0.19 points of the best) while maintaining strong efficiency (8.4% less inference latency).

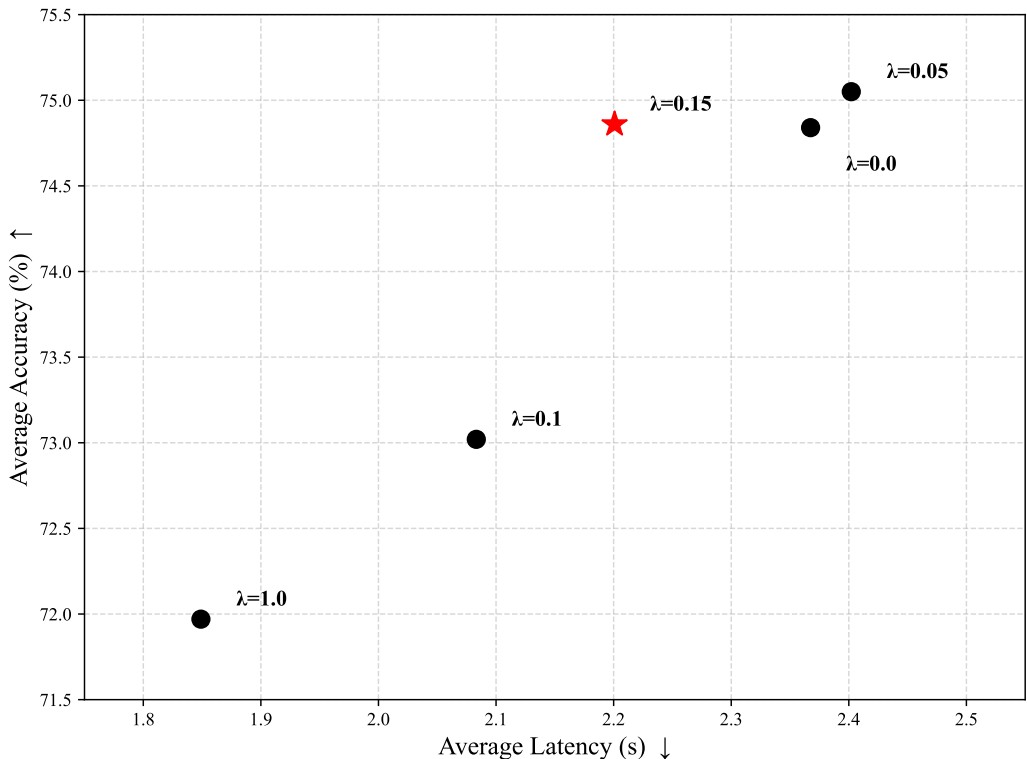

Figure 11: Performance–efficiency trade-off across different values of the resource loss weight ($\lambda$). The red star marks the selected configuration.

Table 11: Inference efficiency for different values of $\lambda$. Higher values encourage more efficient routing, though the effect is not strictly monotonic due to the distinct routing policies induced by each setting.

| $\lambda$ | Latency (s) | Throughput (TPS) |
|---|---|---|
| 0.00 | 2.3675 | 15.14 |
| 0.05 | 2.4021 | 13.72 |
| 0.10 | 2.0832 | 23.85 |
| 0.15 | 2.2008 | 17.77 |
| 1.00 | 1.8490 | 29.56 |

