# OpenReview forum: "TableDART: Dynamic Adaptive Multi-Modal Routing for Table Understanding"
_ICLR.cc/2026/Conference — ICLR 2026 Poster_

### Official Review · Reviewer_P1qR · 2025-10-27

**Soundness:** 2
**Presentation:** 2
**Contribution:** 2
**Rating:** 2
**Confidence:** 4

**Summary:**

This paper proposed TableDart, a method designed for understanding semantic and structual information in table data. It adopted an MLP gating network that dynamically selected the optimal path, avoiding prohibitive costs of full MLLM fine-tuning. It also designed an LLM-based Agent to deal with outputs of other two single-modality experts.

**Strengths:**

- This method only have few parameters to train, avoiding cost of fully finetuning
- The designed dusion agent consider different conditions of other two outputs of single-modality experts, including similar and conflict answers
- This paper do many experiments to validate its performance and efficiency

**Weaknesses:**

There are some core issues that have significant influence on my rating:
- How do the MLP gating network choose the path after giving the logits? Choose the path with the highest score or paths whose scores exceed certain thresholds. If it is the later condition, what if the gating network choose more than one path?
- In **Inference efficiency of section 4.3**, what is the model used in Non-Adaptive baseline? Why the mean inference time can be reduced? From my perspectives, if single-modality expert is chosen, the inference time remain the same. And if the fusion path is chosen, the other two path will be executed, where the inference time will double.
- What's the standard when you choose the single-modality MLLM and the fusion agent as these three models have a significant influence on the performance of your methods.
- Part results in Table 1 are the same with paper "Multimodal Tabular Reasoning with Privileged Structured Information" but no reference is mentioned.

Other discussions:
- Why do you choose three paths, including text-only, image-only and fusion. Why not add another path with MLLM modality? I take the thought that this path will provide more insight of the information than the single-modality experts do.
- I wonder the number of times each path is chosen as this maybe can help understand the inference efficiency.

**Questions:**

Please refer to section Weaknesses.

---

> ### Author Response · Authors · 2025-11-21
>
> ## Response to Weakness 1:
> To clarify, our framework selects the single path corresponding to the **highest output logit** from the gating network. This is achieved via a **standard argmax operation**. This process **guarantees that one and only one path is chosen** for each instance, with ties being resolved deterministically.
> This inference-time argmax decision is **distinct from our training procedure**, where we use a softmax over the logits to form a **soft probability distribution**. This allows the network to learn from multiple potentially correct paths during training while making a single, decisive choice during execution. For better explanation, we have added this explicit clarification to Section 3.3 and the beginning of Section 3.4 in the revised manuscript.
>
> ## Response to Weakness 2:
> To answer your first question, the Non-Adaptive Fusion baseline was designed as a direct ablation study. It uses the exact same set of powerful base models as our full TableDART framework: TableGPT2-7B, Ovis2-8B, and the same LLM-based agent for fusion. The **sole difference lies in the routing strategy**: the Non-Adaptive Fusion baseline adopts a static policy with no routing mechanism, always sending every query–table pair through the full Fusion path. We chose this baseline because it **isolates** the impact of our core contribution, such that any difference in performance or efficiency between TableDART and the baseline can be **attributed solely to the effectiveness of our dynamic routing mechanism**. To further clarify, we have added a more detailed description of the "Non-Adaptive Fusion baseline" in the revised manuscript.
>
> To address your second question, the overall efficiency gain comes from a combination of **dynamic routing and parallel execution**. The **primary** driver of this reduction is **intelligent dynamic routing**. While the inference time for a single instance routed to a unimodal path is standard, the significant drop in the mean time across the whole dataset comes from our policy's ability to avoid the computationally expensive Fusion path for a large majority of cases. The data in Appendix B.3, Figure 10, provides direct evidence: our final model routes **97.2\%** of queries for the TABMWP dataset to the more efficient Image-only path, and **67.5\%** for InfoTabs to the fastest Text-only path. By strategically reserving the costly Fusion path for only the most difficult cases, our framework reduces the average computational cost. Furthermore, your concern about the **Fusion path's latency will double** as the other two paths will be executed is **addressed by parallel execution** (detailed in Appendix A.5.3). When the Fusion path is selected, the two expert models run in parallel, not sequentially. Therefore, the latency is determined by the maximum of their two execution times, not their sum. This prevents the inference time from "doubling," making the Fusion path viable while our intelligent routing ensures it is used only when necessary.
>
> ## Response to Weakness 3:
> We selected these backbone models because they are recent and strong MLLMs within their respective paradigms. We chose TableGPT2-7B for Table-as-Text and Ovis2-8B for Table-as-Image as they were representative state-of-the-art models in the 7-8B class. For the Fusion agent, the criterion was advanced reasoning capabilities to handle the complex "Rescuer" role.
> Crucially, our goal was to demonstrate that our framework provides value on top of these strong models. The results in Table 1 directly validate this. TableDART significantly **outperforms both of its individual backbone models** on average, proving that the **performance gains stem from the effectiveness of our dynamic routing and fusion mechanism**, not just the power of the base models.
> Furthermore, the framework is designed to be **model-agnostic**. This means it can flexibly incorporate other LLMs/MLLMs, including more powerful ones that may be released in the future. The core contribution is the routing framework itself, not the specific instantiation with these two backbone models.
>
> ## Response to Weakness 4:
> Thank you for pointing this out. You are correct that several of the Ovis2 results in Table 1 were originally reported in "Multimodal Tabular Reasoning with Privileged Structured Information" [1]. We appreciate you catching this oversight. These results should indeed have been cited, and we have now **properly added the missing citation** in the revised manuscript.
> To clarify this, we have updated the manuscript (Appendix A.5.2) to: 1) cite the original paper where the results were taken from, 2) explicitly mark which results were re-evaluated in our setting.
> We sincerely thank you for helping us improve the completeness and clarity of our experimental reporting.
>
> ### Reference:
> [1] Jiang, J. P., Xia, Y., Sun, H. L., Lu, S., Chen, ... & Ye, H. J. (2025). Multimodal Tabular Reasoning with Privileged Structured Information. arXiv preprint arXiv:2506.04088.

---

> > ### Author Response · Authors · 2025-11-21
> >
> > ## Response to Other Discussion 1:
> > Our choice of three paths is based on the **fundamental modalities for table understanding**: a text-based path, an image-based path, and a fusion path to synthesize their complementary strengths. This setup allows us to rigorously validate our **core concept: that dynamically routing between these core modalities is superior to a static approach**.
> > While incorporating more expert models is a potential direction for future work, it **introduces a trade-off**. Each additional path increases the system's overhead, including the complexity of the routing decision and the cost of extracting features for the gating network. Our modular framework is designed to support such extensions, but for this foundational work, we focused on demonstrating the effectiveness of the dynamic routing principle with this essential set of paths.
> >
> > ## Response to Other Discussion 2:
> > To clarify this point, Appendix B.3 (Figure 10) of the original transcript provides a detailed breakdown of the **routing distribution** of our model for each dataset.
> > The figure demonstrates that our learned policy is **highly adaptive and does not default to one strategy**. The **path selection frequency varies dramatically based on the dataset's characteristics**. For instance, on the structurally complex TABMWP dataset, where the visual layout is paramount, our model routes **97.2\%** of instances to the more efficient Image-only path. In contrast, for the semantics-focused InfoTabs dataset, the policy shifts to route **67.5\%** of instances to the fastest Text-only path. The powerful but computationally expensive Fusion path is strategically reserved for the most challenging datasets, such as TAT-QA, where single-modality models are far more likely to fail.
> > This data is the direct empirical explanation for the efficiency gains reported in Table 3. The 24.5\% average latency reduction is achieved precisely because our intelligent routing avoids the costly Fusion path for the majority of cases on many datasets.
> >
> > Nevertheless, based on your valuable feedback, we recognize that explicitly linking these results is crucial. Therefore, in our revised manuscript, we have added a sentence at the end of the Inference Efficiency paragraph (Section 4.3), directing the reader to this detailed routing analysis in Appendix B.3, Figure 10, to make this connection clear.

---

> ### Author Response · Authors · 2025-11-27
>
> Dear Reviewer P1qR,
>
> Thank you very much for your valuable time and thoughtful comments. We fully understand that you may have a very busy schedule. As the discussion period has less than one week remaining, we would truly appreciate it if you could kindly take a moment to review our latest responses and let us know whether they adequately address your concerns.
>
> We want to emphasize that our team has dedicated significant time and made every possible effort to carefully address all of your comments. We sincerely hope that this effort is visible to you and that our revisions meet your expectations.
>
> Thank you again for your time and consideration. We would be deeply grateful for your feedback at your earliest convenience.
>
> Best regards,
>
> Authors of Submission 10043

---

> ### Author Response · Authors · 2025-11-28
>
> Dear Reviewer Z7NP and Reviewer P1qR,
>
> As the rebuttal deadline is approaching, I would like to strongly request your timely re-evaluation of our paper in light of the detailed responses we have provided. We have carefully addressed every concern raised and clarified all technical points with substantial supporting evidence.
>
> We also note that the other reviewers have evaluated the work positively, which we believe reflects the merits and contributions of the paper. In this context, we sincerely hope that you two can reconsider your assessment **based on our detailed responses and the revised clarity of the original submission**, with **PROFESSIONALS**.
>
> Thanks.

---

### Official Review · Reviewer_Z7NP · 2025-10-30

**Soundness:** 2
**Presentation:** 2
**Contribution:** 1
**Rating:** 2
**Confidence:** 5

**Summary:**

This paper proposes TableDART, a framework that adaptively selects the optimal processing path for each query to make the most of modality-specific features for table understanding.

**Strengths:**

Proposes a novel dynamic routing framework that adaptively selects the optimal modality (text, image, or fusion) for each table–query pair.

Achieves good performance gains across multiple benchmarks with minimal additional parameters. The parameter-efficient training makes it practical and resource-friendly.

**Weaknesses:**

1. Your results seem insufficient to demonstrate that you achieve Inference Efficiency (the main text only contains a one-sentence description of the results). From the appendix, it can be seen that most tasks on the datasets can be completed with a single modality; however, a fusion strategy is still predominantly used.

2. The two single-modality based methods you used already perform far beyond most models, making many methods seem completely meaningless for comparison. Aren't there any other relatively stronger baselines in this field?

3. Three paths are a bit too few for routing. More models should be adopted to provide more options.

4. How does computational efficiency change as $lambda$ varies? This should be presented clearly. Furthermore, I believe it would be normal for the performance to be optimal when $ \lambda$ is 0, but this does not seem to be the case.

5. From the results, Fusion appears to be the primary source of your performance improvement, and this deserves a detailed explanation.

6. Why not use Qwen2.5-VL or even Qwen3-VL as your baseline?

7. The experimental results in Table 1 on Ovis2 seem partially the same as "Multimodal Tabular Reasoning with Privileged Structured Information" (https://arxiv.org/pdf/2506.04088), except for the result on the TABMWP dataset. However, no reference or explanation can be found in the paper. The authors also did not compare with this paper. Could the authors explain the reason?

**Questions:**

See weakness.

---

> ### Author Response · Authors · 2025-11-21
>
> We sincerely thank you for your time and for providing thoughtful and constructive feedback on our work. The comments have been invaluable in helping us identify areas for improvement.
>
> ## Response to Weakness 1:
> To clarify our claim, the inference efficiency of TableDART is benchmarked against a **Non-Adaptive Fusion baseline**, not against single-modality models. This baseline uses the exact same components of TableDART but is forced to use the expensive fusion path for every instance. As shown in Table 3, our dynamic approach is 24.5\% faster on average, which directly quantifies the efficiency gain from intelligent routing.
> Compared to a single-modality model alone, our framework makes a trade-off. While executing a single-modality path is inherently faster, our framework achieves **higher effectiveness**, as shown by its superior average accuracy in Table 1. This demonstrates a **favorable trade-off** where a potential increase in latency for some queries unlocks a significant gain in overall performance.
>
> Regarding your observation, we respectfully clarify that **not all tasks** can be effectively solved **by a single modality**. The **necessity of Fusion varies significantly across datasets**. For instance, Appendix A.7 (Figure 6) shows that for the challenging TAT-QA dataset, 34.8\% of instances cannot be solved by either single-modality model alone, necessitating Fusion. Consequently, Appendix B.3 Figure 10 shows that our policy adapts by routing **88.7\%** of TAT-QA instances to the Fusion path. Conversely, for datasets where single-modality models suffice, such as TABMWP, the policy avoids Fusion, routing **97.2\%** of instances to the Image-only path. This confirms that our method is **dynamically selective**: it prioritizes efficient single-modality paths when possible and reserves the Fusion path only for complex cases where single modalities fall short. To further clarify this, we have added an explanation in "Section 4.3 Inference Efficiency" to detail why our framework achieves its efficiency improvements.
>
> ## Response to Weakness 2 & Weakness 6:
> Our choice of strong unimodal models was based on the goal of evaluating whether TableDART can provide meaningful improvements even when built on top of the best available models. This allows us to test the framework under a challenging setting where each single-modality expert is already highly capable.
> At the same time, we agree that a broader comparison is important for clearly positioning our method. Following your suggestion, we have expanded our set of baselines. In the revised manuscript, we now **include two additional types of SOTA competitors**:
> - Tabular Domain-specific Table-as-Image Baselines: TabPedia [1] and SynTab-LLaVA [2], to compare against specialized models.
> - A powerful general-purpose Table-as-Image Baseline: Qwen2.5-VL-7B [3], to measure against the latest advancements in general VLMs.
>
> Regarding the Table-as-Multimodality (MLLM-based) category, to the best of our knowledge, there are currently very few open-source works that follow this specific paradigm, with HIPPO [4] being the most prominent example. We believe this underscores the novelty of our approach in this area, and we look forward to comparing TableDART with more such methods as they emerge in the future.
>
> We have incorporated these results into Table 1 of the revised manuscript. Also, we provide the comparison results below.
>
> | Method | WTQ | TABMWP | TAT-QA | HiTab | FeTaQA | TabFact | InfoTabs | Average |
> | :--- | :---: | :---: | :---: | :---: | :---: | :---: | :---: | :---: |
> | TabPedia-7B | 22.53 | 10.66 | 13.08 | 6.54 | 14.31 | 35.49 | 2.43 | 17.78 |
> | SynTab-LLaVA-7B | 39.59 | **88.30** | 51.94 | 35.66 | 35.45 | 70.78 | 69.42 | 59.28 |
> | Qwen2.5-VL-7B | 54.37 | 63.69 | 51.94 | 62.69 | 10.99 | 75.81 | 70.13 | 63.11 |
> | **TableDART (TableGPT2-7B + Ovis2-8B)** | **70.58** | 84.54 | **62.05** | **74.37** | **36.11** | **81.37** | **76.22** | **74.86** |
>
> ### Reference:
> [1] Zhao, W., Feng, H., Liu, Q., Tang, J., Wu, B., Liao, L., ... & Huang, C. (2024). Tabpedia: Towards comprehensive visual table understanding with concept synergy. Advances in Neural Information Processing Systems, 37, 7185-7212.
>
> [2] Zhou, B., Gao, Z., Wang, Z., Zhang, B., Wang, Y., Chen, Z., & Xie, H. (2025). SynTab-LLaVA: Enhancing Multimodal Table Understanding with Decoupled Synthesis. In Proceedings of the Computer Vision and Pattern Recognition Conference (pp. 24796-24806).
>
> [3] Bai, S., Chen, K., Liu, X., Wang, J., Ge, W., Song, S., ... & Lin, J. (2025). Qwen2. 5-vl technical report. arXiv preprint arXiv:2502.13923.
>
> [4] Liu, Z., Wang, H., Li, X., Xiong, Q., Yang, X., Gu, Y., ... & Sun, M. (2025). Hippo: Enhancing the table understanding capability of large language models through hybrid-modal preference optimization. arXiv preprint arXiv:2502.17315.

---

> > ### Author Response · Authors · 2025-11-21
> >
> > ## Response to Weakness 3:
> > Our choice of three paths is **based on the fundamental modalities for table understanding**: a text-based path, an image-based path, and a fusion path to synthesize their complementary strengths. This setup allows us to rigorously validate our **core concept: that dynamically routing between these core modalities is superior to a static approach.**
> > While incorporating more expert models may potentially improve the performance, it introduces more trade-offs for computational costs and system complexity. Our modular framework is designed to support such extensions, but for this foundational work, we focused on demonstrating the effectiveness of the dynamic routing principle with this essential set of paths.
> >
> > ## Response to Weakness 4:
> > A key role of $\lambda$ is to act as an effective **regularizer that improves generalization**.
> > The relationship between $\lambda$ and efficiency is inherently **non-linear**, driven by the three-path trade-off learned by the routing policy.
> >
> > To provide a clearer, data-driven explanation, we have conducted a **new detailed analysis**. In the revised manuscript, we have added additional discussion in "Section 4.5 — Impact of Resource-Aware Objective", included a new Appendix Section (Appendix B.4), and also a figure (Figure 11) that illustrates **how varying  $\lambda$ affects efficiency**. Specifically, the figure plots measured average latency on the $y$-axis against the corresponding average accuracy (from Table 5) on the $x$-axis across different $\lambda$ values.
> >  The full numerical results, including both latency and throughput (TPS), are now included in a new appendix table (Table 11) for complete transparency.
> > This new visualization directly addresses both parts of your question:
> >
> > - First, regarding how efficiency changes with $\lambda$, Figure 11 shows that **although a higher $\lambda$ generally encourages more efficient behavior**, the relationship is **not strictly monotonic**. For example, the policy trained with $\lambda=0.1$ exhibits lower average latency than the one trained with $\lambda=0.15$. This is a critical insight: each value of $\lambda$ induces a distinct routing policy, and the policy at $\lambda=0.15$ is more effective at identifying the truly difficult queries that require the slower Fusion path to be answered correctly. This yields higher accuracy but also increases latency. In contrast, the $\lambda=0.1$ policy routes more of these hard cases to faster but incorrect unimodal paths, which reduces latency at the cost of a noticeable performance drop. This explains the non-monotonicity and reflects the nuanced interaction among the three expert paths.
> >
> > - Second, this directly explains **why the configuration with  $\lambda = 0$ is not optimal in terms of accuracy**. Our new figure shows that $\lambda=0.15$ achieves the highest average accuracy and lies on the optimal region of the performance–efficiency frontier. Importantly, it is not only more accurate but also more efficient (lower latency) than both $\lambda=0.0$ and $\lambda=0.05$. This demonstrates the **core principle of our design: the resource-aware objective regularizes the routing policy**. A policy trained with $\lambda=0$ focuses exclusively on maximizing training-set performance, leading to over-reliance on the expensive Fusion path and poorer generalization. Introducing a small resource penalty encourages the model to learn a more **balanced and better-generalized** routing strategy.
> >
> > ## Response to Weakness 5:
> > While our powerful Fusion agent is indeed essential for resolving complex cases, the primary driver of TableDART’s superior performance and efficiency is **not the fusion agent itself**, but the **intelligent decision of when to invoke it**.
> > This can be evident by the ablation study in Section 4.5 (Table 4), where we compare our full TableDART framework (with dynamic routing) against a *TableDART w/ Non-Adaptive Fusion* baseline. This compared baseline employs the exact same powerful Fusion agent, but is forced to apply it to every single instance—completely removing the routing mechanism.
> > Compared with always using the Fusion agent regardless of input, our dynamic routing approach achieves **superior performance** on multiple datasets (e.g., +3.07\% on TABMWP and +1.02\% on HiTab) and remains highly competitive on the others, while being substantially more efficient (as shown in Section 4.2 Table 2). This clearly demonstrates that it is the routing mechanism itself,  rather than simply the presence of a strong Fusion agent, that drives the overall success of the framework.
> > To further clarify this point, we have added an explanation at the end of "Section 4.5 — Impact of Dynamic Gating" in the revised manuscript, highlighting that the performance gains arise not only from the powerful Fusion but also, critically, from the dynamic routing mechanism.

---

> > > ### Author Response · Authors · 2025-11-21
> > >
> > > ## Response to Weakness 7:
> > > Thank you for pointing this out. You are correct that several of the Ovis2 results in Table 1 were originally reported in "Multimodal Tabular Reasoning with Privileged Structured Information" [1]. We appreciate you catching this oversight. These results should indeed have been cited, and we have now **properly added the missing citation** in the revised manuscript.
> > >
> > > - Regarding **why this work was not included as a main baseline**: the method enhances a single vision–language model using privileged structured annotations that are available only during training. This paradigm is conceptually orthogonal to our setting. TableDART is a multi-modal dynamic routing framework that does not rely on privileged information and instead focuses on optimizing the performance–efficiency trade-off by deciding when to invoke different pretrained LLM models. Because the privileged-information method modifies a single multimodal model rather than selecting among different models, it does not constitute a directly comparable baseline for our framework.
> > >
> > > - Regarding the discrepancy on the TABMWP dataset: under our experimental pipeline, we were unable to reproduce the exact TABMWP result reported in that paper, despite our best efforts to follow all publicly available hyperparameters and preprocessing steps as closely as possible. To maintain fairness and ensure consistency across all baselines, we therefore re-evaluated Ovis2 under our unified experimental pipeline on TABMWP, and the value reported in Table 1 corresponds to this newly obtained result.
> > >
> > > To clarify this, we have updated the manuscript (Appendix A.5.2) to: 1) cite the original paper where the results were taken from, 2) explicitly mark which results were re-evaluated in our setting, and 3) explain the reason for the discrepancy on TABMWP.
> > >
> > > We sincerely thank you for helping us improve the completeness and clarity of our experimental reporting.
> > >
> > > ### Reference:
> > > [1] Jiang, J. P., Xia, Y., Sun, H. L., Lu, S., Chen, Q. G., Luo, W., ... & Ye, H. J. (2025). Multimodal Tabular Reasoning with Privileged Structured Information. arXiv preprint arXiv:2506.04088.

---

> > > > ### Comment · Reviewer_Z7NP · 2025-11-24
> > > >
> > > > Thank you for the rebuttal. While the dynamic routing concept is promising, my concerns persist:
> > > >
> > > > Routing’s Potential vs. Reality: The framework’s adaptability is conceptually appealing, but performance gains are inconsistent.
> > > >
> > > > Unclear Modality Logic: The routing criteria (e.g., task "difficulty") remain poorly defined. Why does fusion dominate TAT-QA but not TABMWP? Concrete links or explanations to table/query attributes are missing. More analysis on what to route and when to route will be beneficial.
> > > >
> > > > Training Efficiency: While parameter-efficient, the framework still trains two large models (~7B each). The reported 13.5h training time is not trivial, especially compared to end-to-end methods such as Turbo that may amortize costs over a single model.
> > > >
> > > > Inference Efficiency: The 24.5% speedup over Non-Adaptive Fusion is meaningful, but this comparison is skewed. A single-modality model (e.g., Ovis2-8B) would inherently outperform TableDART in latency. The trade-off between accuracy and latency remains suboptimal for real-world deployment. Even with routing, the fusion path introduces unavoidable computational overhead.
> > > >
> > > > According to these concerns, I will maintain my original rating.

---

> > > > > ### Author Response · Authors · 2025-11-24
> > > > >
> > > > > Thanks for the reviewer's timely reply. The following is our response to your remaining concerns.
> > > > > ## Response to Routing’s Potential vs. Reality:
> > > > > The "consistency" of our framework is demonstrated by two key facts in Table 1:
> > > > > - **It consistently outperforms its own components.** The TableDART (Ovis2 variant) achieves an average accuracy of 74.86\%, which is a significant and consistent improvement of over 5\% compared to both its Table-as-Text model (69.14\%) and its Table-as-Image model (69.49\%). This proves that the dynamic routing and fusion mechanism consistently adds substantial value.
> > > > > - **It achieves a top-tier average performance.** Across all seven benchmarks, our framework achieves the highest average accuracy among all compared methods. This demonstrates its robustness and effectiveness as a general-purpose table understanding framework, which is the primary claim of our work.
> > > > >
> > > > > ## Response to Unclear Modality Logic:
> > > > > The routing logic is not based on predefined "difficulty" rules but is a direct, learned response to the **inherent characteristics of the data** in each benchmark. For instance, the routing differences between TAT-QA and TABMWP are a clear demonstration of this:
> > > > > - **TAT-QA** is known for hybrid questions that require complex reasoning over both the table and associated free-form text paragraphs. This makes it inherently difficult for any single modality to succeed alone. A text-only view can struggle with complex table structures, while an image-only view may fail to integrate the crucial and precise information from the separate text. This inherent need for both views is why our empirical analysis (Appendix A.7, Figure 6) shows that **34.8\% of its samples are unsolvable by either single-modality models alone**. Consequently, the gating network correctly learns that fusion is the optimal strategy and routes 88.7\% of TAT-QA cases to the powerful Fusion path (Appendix B.3, Figure 10).
> > > > > - **TABMWP**, in contrast, consists of math problems where preserving the table's visual structure and numerical layout is critical for correctly setting up calculations. Serializing such tables into a flat text string can disrupt these crucial spatial relationships. Recent work indicates that image-based representations can outperform text serialization on complex numerical reasoning tasks [1], suggesting that preserving visual structure can provide advantages for similar numerical reasoning TABMWP. Our analysis (Appendix A.7, Figure 6) confirms that both single-modality models are highly effective for TABMWP while the Image-only path demonstrates a higher overall success rate. Therefore, the gating network learns this and routes 97.2\% of cases to the Image-only path.
> > > > >
> > > > > This demonstrates that the routing logic is a direct, learned response to the **data's fundamental characteristics**. The gating network correctly identifies when a task requires integrating multiple sources (necessitating fusion), when a task relies on fine-grained semantics (favoring a text-based view), or when a task is structurally-driven (favoring an image-based view).
> > > > >
> > > > > ## Response to Training Efficiency:
> > > > > There is a **misunderstanding** regarding our training procedure. We do not train two large models. The 7B/8B-scale backbone models are kept **completely frozen at all times**; their weights are never updated. The only component that is trained is the lightweight, **2.59M-parameter** MLP gating network. Also, a direct comparison of training time is not meaningful. The most relevant metric is **the number of trainable parameters**. Our method requires training only 2.59M parameters, which is orders of magnitude smaller than the end-to-end approach that involves fine-tuning a 7B-scale model. The comparison to Turbo [2] is also not applicable. Turbo's aim is to enhance a single vision-language model by leveraging structured tables as privileged information during training. **Our work addresses a different problem:** dynamically selecting between text-only, image-only, or their fusion. These are orthogonal research questions.
> > > > >
> > > > > ### Reference:
> > > > > [1] Deng, N., Sun, Z., He, R., Sikka, A., Chen, Y., Ma, L., … Mihalcea, R. (2024, August). Tables as Texts or Images: Evaluating the Table Reasoning Ability of LLMs and MLLMs. In L.-W. Ku, A. Martins, & V. Srikumar (Eds), Findings of the Association for Computational Linguistics: ACL 2024 (pp. 407–426). doi:10.18653/v1/2024.findings-acl.23
> > > > >
> > > > > [2] Jiang, J. P., Xia, Y., Sun, H. L., Lu, S., Chen, Q. G., Luo, W., ... & Ye, H. J. (2025). Multimodal Tabular Reasoning with Privileged Structured Information. arXiv preprint arXiv:2506.04088.

---

> > > > > > ### Author Response · Authors · 2025-11-24
> > > > > >
> > > > > > ## Response to Inference Efficiency:
> > > > > > - The latency comparison to a single-modality model is not a fair benchmark for a dynamic routing framework. Our system inherently utilizes two expert models and a fusion agent to achieve higher accuracy, so it is expected to have higher latency than a single model. Our goal is to **improve effectiveness while intelligently managing efficiency**. Therefore, to isolate the value of the routing mechanism itself, the fair comparison is against the Non-Adaptive Fusion baseline. Our 24.5\% speedup in this direct comparison is not skewed; it is the precise, empirical measure of the efficiency gained by our intelligent routing logic.
> > > > > > - Furthermore, the claim that our accuracy-latency trade-off is "suboptimal" is inaccurate. Our framework provides a demonstrably superior trade-off as evidenced by our results: 1) Compared to its *constituent backbone models*, TableDART is significantly more effective. It achieves an average accuracy of 74.86\%, surpassing both the Text-only path (69.14\%) and the Image-only path (69.49\%). 2) Compared to the *Non-Adaptive Fusion baseline*, TableDART is not only 24.5\% more efficient (Table 3) but also achieves a higher average accuracy (74.86\% vs. 74.59\%) and outperforms it on 3 out of 7 datasets (Table 4).
> > > > > > - Finally, the "unavoidable computational overhead" of the fusion path is, in fact, avoided by our framework in the majority of cases. The entire purpose and demonstrated outcome of our routing mechanism is to reserve this overhead for only the most difficult instances where it is empirically necessary to improve accuracy.

---

> ### Author Response · Authors · 2025-11-27
>
> Dear Reviewer Z7NP,
>
> Thank you very much for your valuable time and thoughtful comments. We fully understand that you may have a very busy schedule. As the discussion period has less than one week remaining, we would truly appreciate it if you could kindly take a moment to review our latest responses and let us know whether they adequately address your concerns.
>
> We want to emphasize that our team has dedicated significant time and made every possible effort to carefully address all of your comments. We sincerely hope that this effort is visible to you and that our revisions meet your expectations.
>
> Thank you again for your time and consideration. We would be deeply grateful for your feedback at your earliest convenience.
>
> Best regards,
>
> Authors of Submission 10043

---

> ### Author Response · Authors · 2025-11-28
>
> Dear Reviewer Z7NP and Reviewer P1qR,
>
> As the rebuttal deadline is approaching, I would like to strongly request your timely re-evaluation of our paper in light of the detailed responses we have provided. We have carefully addressed every concern raised and clarified all technical points with substantial supporting evidence.
>
> We also note that the other reviewers have evaluated the work positively, which we believe reflects the merits and contributions of the paper. In this context, we sincerely hope that you will reconsider your assessment **based on our detailed responses and the revised clarity of the original submission**, with **PROFESSIONALS**.
>
> Thanks.

---

### Official Review · Reviewer_nQLA · 2025-10-31

**Soundness:** 3
**Presentation:** 2
**Contribution:** 3
**Rating:** 4
**Confidence:** 4

**Summary:**

This paper proposes TableDART—a training-efficient framework that reuses pretrained single-modality models via a lightweight 2.59M-parameter MLP gating network, dynamically selecting the optimal path (Text-only, Image-only, or Fusion) for each table–query pair to mitigate redundancy and conflicts.

**Strengths:**

TableDART addresses limitations of existing table understanding methods by introducing a training-efficient framework with a lightweight MLP gating network, dynamically selecting optimal (text, image, or fusion) paths per table-query pair to reduce redundancy/conflicts. It further incorporates a cross-modal agent for knowledge integration without costly MLLM fine-tuning, achieving SOTA on seven benchmarks.

**Weaknesses:**

1. The paper is difficult to comprehend, as many concepts are employed directly without any corresponding explanation of their meanings—
   - For instance, what does the term "fine-grained" on line 46 specifically refer to?
   - As noted in the abstract: "statically process both modalities for every query-table pair within large multimodal LLMs (MLLMs), inevitably introducing redundancy and even conflicts," what exactly does the term "conflicts" refer to herein?
   - In the Introduction, it is mentioned that table data exhibits the property of heterogeneity—could the authors provide an explanation for this?
   - What modality exactly is Table T on line 145—textual or visual?
2. The literature review is inadequate, and the compared methods lack several representative approaches, such as the domain-specific VLM TabPedia、SynTab-LLaVA, and the general VLM Qwen2.5VL.
3. How are the three token sequences (e_q, e_t, e_v), which vary in length and differ in token dimension, concatenated?
4. As the authors state, "We employ TableGPT2-7B and Ovis2-8B as TableDART’s single-modality experts," it is queried whether ablation experiments have been conducted using other alternative single-modality experts to verify the robustness and generalizability of the proposed framework.

**Questions:**

see the weakness

---

> ### Author Response · Authors · 2025-11-21
>
> Thank you for your time and valuable feedback. Your comments have greatly helped us improve our work.
> ## Response to Weakness 1:
> We recognize the importance of defining our terminology precisely, and we have provided clearer explanations below. Also, we have incorporated these clarifications into the revised manuscript to improve overall clarity.
> - **"Fine-grained semantics":** This term refers to the **precise and nuanced** meaning embedded within the text of a table's cells and headers. For example, understanding the subtle but critical difference between "quarterly revenue" and "annual profit" requires a deep linguistic comprehension that goes beyond simply recognizing words. While a vision-based model can capture a table's overall structure and perform optical character recognition (OCR) to extract the text, it may not possess the same level of deep semantic understanding as a large language model specifically trained on tables with text representation. Our work addresses the challenge of choosing the right modality when such fine-grained understanding is critical.
> - **"Conflicts":** This refers to **contradictory signals** that can arise when a model is forced to process both the textual and visual representations of a table simultaneously. A prime example, as mentioned in our introduction, is the issue of permutation invariance. A textual serialization of a table (Table-as-Text) inevitably presents the rows in a fixed, linear order. A language model might incorrectly infer that this order is meaningful. In contrast, the table's visual representation (Table-as-Image) is permutation-invariant with respect to its rows. A multimodal model statically processing both views is therefore fed conflicting cues: one suggesting row order matters, and the other suggesting it does not. This internal contradiction can confuse the model's reasoning process, leading to errors. Our dynamic routing avoids such conflicts by selecting only the most appropriate view(s).
> - **"Heterogeneity":** This is a fundamental property of tabular data, referring to the fact that a single table typically contains **columns with a wide variety of data types**. For instance, a table about products might have a 'Product ID' column (categorical), a 'Price' column (numerical), a `Description' column (free-form text), and a 'Launch Date' column (temporal). This mix of data types makes tables inherently challenging to model, as a single, uniform processing strategy is often suboptimal. Effectively capturing the information from each column type requires handling this diversity.
> - **"Table T":** This notation refers to the original, underlying table data as a **raw, unprocessed data structure**. As shown in our methodology (Eq. 1), this abstract 'T' is then passed to two separate functions to create the inputs for our models: 'Serialize(T)' converts it into a linearized text sequence for the text-based model, while 'Screenshot(T)' renders it as a pixel-based image for the vision-language model.
>
> ## Response to Weakness 2:
> To provide a more comprehensive and robust evaluation, we have taken action on two fronts.
> First, we have now **included the tabular domain-specific SOTA VLMs**, TabPedia [1] and SynTab-LLaVA [2], as well as the strong general-purpose VLM Qwen2.5-VL-7B [3], in our main baseline comparison in Table 1.
> Second, in our revised manuscript, we have **expanded the Related Work section** to incorporate these relevant works. Specifically, within our discussion of the Table-as-Image paradigm in Section 2, we have added a detailed discussion of these important contributions, properly situating our framework with respect to recent advances in both general and domain-specific table understanding models. Below, we provide the corresponding comparison results.
> | Method | WTQ | TABMWP | TAT-QA | HiTab | FeTaQA | TabFact | InfoTabs | Average |
> | :--- | :---: | :---: | :---: | :---: | :---: | :---: | :---: | :---: |
> | TabPedia-7B | 23.53 | 10.66 | 13.08 | 6.54 | 14.31 | 35.49 | 2.43 | 15.29 |
> | SynTab-LLaVA-7B | 39.59 | **88.30** | 51.94 | 35.66 | 35.45 | 70.78 | 69.42 | 59.28 |
> | Qwen2.5-VL-7B | 54.37 | 63.69 | 51.94 | 62.69 | 10.99 | 75.81 | 70.13 | 63.11 |
> | **TableDART (TableGPT2-7B + Ovis2-8B)** | **70.58** | 84.54 | **62.05** | **74.37** | **36.11** | **81.37** | **76.22** | **74.86** |
>
> ### Reference:
>
> [1] Zhao, W., Feng, H., Liu, Q., Tang, J., Wu, B., Liao, L., ... & Huang, C. (2024). Tabpedia: Towards comprehensive visual table understanding with concept synergy. Advances in Neural Information Processing Systems, 37, 7185-7212.
>
> [2] Zhou, B., Gao, Z., Wang, Z., Zhang, B., Wang, Y., Chen, Z., & Xie, H. (2025). SynTab-LLaVA: Enhancing Multimodal Table Understanding with Decoupled Synthesis. In Proceedings of the Computer Vision and Pattern Recognition Conference (pp. 24796-24806).
>
> [3] Bai, S., Chen, K., Liu, X., Wang, J., Ge, W., Song, S., ... & Lin, J. (2025). Qwen2. 5-vl technical report. arXiv preprint arXiv:2502.13923.

---

> > ### Author Response · Authors · 2025-11-21
> >
> > ## Response to Weakness 3:
> > - To clarify, we **do not concatenate the variable-length token sequences directly**. Instead, as detailed with full specifications in our Appendix A.3.1, we first employ **modality-specific pooling** operations to convert the variable-length outputs from each encoder into fixed-dimensional feature vectors. It is these resulting fixed-size vectors that are then concatenated.
> > - To avoid any ambiguity in the revised manuscript, the sentence on line 151 of the original transcript, which currently reads: "The resulting feature vectors are concatenated to form a multimodal representation of the table-query pair:"
> > have been **revised** to be more explicit:
> > "To form a unified, fixed-size input for the gating network, we first apply modality-specific pooling strategies to these encoder outputs, converting them into fixed-dimensional feature vectors. These resulting vectors are then concatenated to form a multimodal representation of the table-query pair:".
> >
> > ## Response to Weakness 4:
> > To verify the robustness and generalizability of our framework, we conducted an **additional ablation experiment** by **replacing the Table-as-Image model** (Ovis2-8B) with **Qwen2.5-VL-7B** [1], a state-of-the-art general-purpose VLM, while keeping the Text expert (TableGPT2-7B) fixed.
> > The results, presented in the table below, demonstrate the strong **model-agnostic capability** of TableDART. When instantiated with Qwen2.5-VL-7B, our framework achieves an average accuracy of 70.25\%, representing a **+7.14\%** gain over the single-modality Qwen2.5-VL baseline (63.11\%). Notably, this variant also achieves competitive performance compared to the MLLM-based baseline HIPPO-8B (70.84\%) and significantly outperforms Gemini 2.0 Flash (56.92\%). These findings confirm that the effectiveness of TableDART stems from its **dynamic routing mechanism** rather than being tied to specific base models. We have also incorporated these results into Table 1 of the revised manuscript.
> >
> > | Method | WTQ | TABMWP | TAT-QA | HiTab | FeTaQA | TabFact | InfoTabs | Average |
> > | :--- | :---: | :---: | :---: | :---: | :---: | :---: | :---: | :---: |
> > | *Constituent Models* | | | | | | | | |
> > | TableGPT2-7B *(Text-only Path)* | 61.42 | 83.87 | 50.39 | 70.27 | 28.97 | 77.80 | 71.07 | 69.14 |
> > | Ovis2-8B *(Image-only Path)* | 58.76 | 87.00 | 47.67 | 68.59 | 34.70 | 80.80 | 74.11 | 69.49 |
> > | Qwen2.5-VL-7B *(Image-only Path)* | 54.37 | 63.69 | 51.94 | 62.69 | 10.99 | 75.81 | 70.13 | 63.11 |
> > | *MLLM-based Baselines* | | | | | | | | |
> > | HIPPO-8B | 55.77 | **87.50** | 60.75 | 63.00 | 33.18 | 82.27 | 75.74 | 70.84 |
> > | Gemini 2.0 Flash | 63.56 | 46.29 | 35.62 | 60.41 | 10.57 | 81.33 | 54.31 | 56.92 |
> > | *Dynamic Adaptive Routing* | | | | | | | | |
> > | **TableDART (TabGPT2 + Ovis2)** | **70.58** | 84.54 | **62.05** | **74.37** | **36.11** | **81.37** | **76.22** | **74.86** |
> > | TableDART (TableGPT2 + Qwen2.5VL) | 69.29 | 72.61 | 59.07 | 71.13 | 29.87 | 77.94 | 71.46 | 70.25 |
> >
> > ### Reference:
> > [1] Bai, S., Chen, K., Liu, X., Wang, J., Ge, W., Song, S., ... & Lin, J. (2025). Qwen2. 5-vl technical report. arXiv preprint arXiv:2502.13923.

---

> ### Author Response · Authors · 2025-11-27
>
> Dear Reviewer nQLA,
>
> Thank you very much for your valuable time and thoughtful comments. We fully understand that you may have a very busy schedule. As the discussion period has less than one week remaining, we would truly appreciate it if you could kindly take a moment to review our latest responses and let us know whether they adequately address your concerns.
>
> We want to emphasize that our team has dedicated significant time and made every possible effort to carefully address all of your comments. We sincerely hope that this effort is visible to you and that our revisions meet your expectations.
>
> Thank you again for your time and consideration. We would be deeply grateful for your feedback at your earliest convenience.
>
> Best regards,
>
> Authors of Submission 10043

---

> > ### Comment · Reviewer_nQLA · 2025-11-28
> >
> > Thank you for your response and the experimental results provided. My previous concerns have been adequately addressed. I will provide a positive rating.

---

### Official Review · Reviewer_YExZ · 2025-11-04

**Soundness:** 3
**Presentation:** 3
**Contribution:** 3
**Rating:** 8
**Confidence:** 3

**Summary:**

The paper proposes TableDART, a dynamic adaptive multi-modal routing framework for table understanding tasks. The key innovation lies in replacing traditional monolithic multimodal LLM fine-tuning with a lightweight 2.59M-parameter gating network that dynamically selects among three inference paths, Text-only, Image-only, or Fusion, depending on each table–query pair’s characteristics. The fusion stage employs an LLM agent that arbitrates between unimodal outputs or synthesizes a new answer when both fail. TableDART reuses frozen pretrained unimodal models, drastically reducing training cost. Experiments on seven benchmarks show TableDART surpasses strong baselines.

**Strengths:**

- The dynamic adaptive routing approach is impressive. It represents a conceptual shift from static multimodal fusion to instance-specific modality selection.

- The paper demonstrates strong empirical rigor.

- TableDART addresses a key bottleneck in multimodal table understanding, redundant modality processing and high fine-tuning cost

**Weaknesses:**

- While the paper highlights the LLM-based Fusion agent's "Arbitrator" and "Rescuer" roles, its decision logic is black-box. There is no systematic evaluation of when or why it succeeds beyond anecdotal cases.

- The use of Gemini 2.0 Flash for fusion raises reproducibility and accessibility concerns. A variant using a fully open LLM would improve transparency.

- The gating network's supervision relies on pre-computed correctness vectors, but the procedure for labeled paths is not critically examined.

- It would be instructive to show a variant where fusion uses a simple weighted ensemble rather than an LLM agent, to disentangle the benefits of adaptive routing at the fusion stage.

**Questions:**

- How sensitive is the routing policy to the composition of the training mixture? Would domain-specific fine-tuning alter routing distributions?

- Does the gating network rely primarily on query semantics or table structure cues when choosing between text and image paths?

---

> ### Author Response · Authors · 2025-11-21
>
> We sincerely thank you for your time and for providing thoughtful and constructive feedback on our manuscript. The comments have been invaluable in helping us identify areas for improvement.
>
> ## Weakness 1 Response:
> To systematically evaluate the Fusion agent's "Arbitrator" and "Rescuer" roles beyond the case studies, we analyzed its performance across the entire test set using two key metrics, which are visualized in Figure 2b of our paper.
> - To evaluate the **"Arbitrator"** role, we first identified when it is needed. This occurs in cases where the two single-modality models produce conflicting outputs (one correct, one incorrect). Our **"Complementarity Rate" metric quantifies** this phenomenon, showing that such conflicts are common, occurring in over 20\% of cases on datasets like WTQ. The Fusion agent succeeds in these scenarios because it can effectively assess the reasoning paths of both experts against the source table and select the more reliable answer, which is a key reason for our overall performance lift.
> - To evaluate the **"Rescuer"** role, we focused on the most difficult cases: when both single-modality models fail. Our **"Synergy Success Rate" metric** directly measures the Fusion agent's success rate in these specific situations. The results show that, on average, the agent successfully resolves 14.0\% of these "hard cases". It succeeds because it is not just choosing between two options, but is capable of synthesizing a new, correct answer by **combining partially correct fragments** of information from both of the failed outputs, as illustrated in our case study (Figure 4). This systematic success in otherwise unsolvable situations provides strong quantitative evidence for its advanced reasoning capabilities.
>
> ## Weakness 2 Response:
> To ensure the reproducibility of our results, we implemented two measures.
> 1) We used the **stable version** of Gemini 2.0 Flash, whose latest update is documented as February 2025 in the official Google API specification, and configured the API to produce fully **deterministic outputs** to remove sampling randomness.
> 2) We provide our checkpoint and code, including all configuration details, in the anonymous GitHub repository.
> While Gemini was selected to evaluate the upper bound of our framework under a strong reasoning model, a key strength of TableDART is its **plug-and-play modularity**. The core contribution lies in the dynamic routing framework, which is **model-agnostic** and can readily incorporate any LLM, ensuring its long-term value and adaptability for the research community.
>
> ## Weakness 3 Response:
> The supervision for our gating network is entirely empirical, based on the actual performance of each path.
> For each training sample, we first determine a correctness vector by **checking which paths (Text, Image, or Fusion) produce the correct answer**. Crucially, if multiple paths are successful, we do not arbitrarily pick one as the "best". Instead, we convert this vector into a soft probability distribution using a temperature-controlled softmax. The gating network is then trained to match this soft target by minimizing KL Divergence. This approach teaches the network to recognize all viable options, avoiding a fragile policy.
> The evidence of this strategy is the policy's **strong generalization performance**, as shown in the zero-shot experiments in Section 4.2 Table 2. Our model's average accuracy remains nearly unchanged, **dropping by less than 1\%** from 74.95\% on seen datasets to 74.37\% on unseen ones. In contrast, the strong MLLM baseline HIPPO-8B suffers a significant performance degradation, dropping from 72.41\% to 63.00\% on the same split. This result provides empirical evidence that our network learned a robust and generalizable routing strategy from the supervision, rather than memorizing patterns specific to the training set.

---

> > ### Author Response · Authors · 2025-11-21
> >
> > ## Weakness 4 Response:
> > We agree that comparing with a simpler fusion mechanism is valuable. However, a simple weighted ensemble is not well-suited for our task due to the heterogeneous outputs (e.g., free-form text, numerical values) where mechanisms like voting are not well-defined. For this reason, an LLM agent with strong reasoning ability is needed to fuse the complex outputs, which is why we adopt it as our fusion module.
> >
> > To directly address your question about disentangling the benefits of the routing mechanism itself, we refer to the ablation study in Section 4.5 Table 4. We compare our framework against a *Random Routing baseline* and a *Non-Adaptive Fusion baseline* (same backbone models as TableDART, but uses the same Fusion agent for all instances instead of dynamic routing). The results demonstrate that TableDART **consistently outperforms** Random Routing, confirming the validity of the learned policy. Furthermore, TableDART achieves **comparable or superior performance** to the Non-Adaptive Fusion baseline, notably surpassing it on datasets like TABMWP (+3.07\%) and HiTab (+1.02\%). This empirical evidence confirms that the framework's effectiveness stems from the **intelligent routing mechanism** minimizing redundancy and conflicts, rather than solely from the capabilities of the fusion agent.
> >
> > ## Question 1 Response:
> > To address your first question, our experiments show that the learned policy is robust to the training mixture. The zero-shot generalization results in Section 4.2 (Table 2) provide the evidence: the framework’s average accuracy on **unseen** datasets (74.37\%) is **nearly identical to** its accuracy on **seen** datasets (74.95\%). This indicates that the network has learned a generalizable routing skill rather than overfitting to the training distribution.
> >
> > Regarding your second question, we believe that domain-specific fine-tuning **would indeed alter the routing distributions**. Our current model is trained on five diverse datasets, which allows the gating network to acquire a broadly applicable routing ability for table understanding. When adapted to a specific domain, the policy would naturally shift to reflect the characteristics of that domain. For example, it would favor the Text-only path for finance report tables rich with specific terminology, or the Image-only path for structurally complex scientific tables.
> > Importantly, we view this domain-specific adaptability as a strength of the framework. Because TableDART is model-agnostic, practitioners can integrate domain-specific expert models and fine-tune only the lightweight (2.59M parameter) gating network. This makes domain adaptation efficient and highlights the framework’s flexibility for practical applications.
> >
> > ## Question 2 Response:
> > The gating network uses **both** query semantics and table structure cues **depending on the specific input**, instead of always preferring one over the other. As evidence, the analysis of our final model’s routing policy in Appendix B.3 (Figure 10) shows this adaptive behavior.
> > For example, on the TABMWP dataset, which requires complex structural reasoning, the policy routes 97.2\% of instances to the Image-only path, indicating it prioritizes table structure cues. In contrast, for the InfoTabs dataset, a task demanding precise semantic understanding, the policy routes 67.5\% of instances to the Text-only path, showing it prioritizes query semantics and textual content. This behavior confirms that the network has learned to arbitrate between these different cues to make the most effective routing decision for each input case.

---

> > > ### Comment · Reviewer_YExZ · 2025-11-21
> > >
> > > Thanks for the reply and there is no further questions. I will maintain my positive rating.

---

### Author Response · Authors · 2025-11-21
**Summary of Manuscript Revisions**

We sincerely thank all reviewers for the insightful review and detailed comments. To address the concerns, we have made the following revisions to the manuscript:
1. Verified the generalizability of TableDART by conducting additional experiments with a variant instantiated using Qwen2.5-VL-7B as the Table-as-Image model. We have updated the main results in `Table 1` and added a discussion on model-agnostic effectiveness to `Section 4.2`.
2. Expanded the baseline comparison to include tabular domain-specific SOTA models (TabPedia, SynTab-LLaVA) and the general-purpose SOTA VLM Qwen2.5-VL-7B. These results have been incorporated into `Table 1`.
3. Deepened the efficiency analysis in `Section 4.3`. We clarified the definition of the Non-Adaptive Fusion baseline and explicitly analyzed how the routing distribution (e.g., avoiding Fusion for simple queries) drives efficiency gains. Furthermore, we added a new analysis on the trade-off between the resource-aware objective ($\lambda$) and inference latency in `Section 4.5` and `Appendix B.4`.
4. Expanded the literature review in `Section 2` to summarize more recent Table-as-Image advancements, specifically incorporating discussions on TabPedia, SynTab-LLaVA, and Qwen2.5-VL.
5. Provided a detailed rationale for our model selection choices in `Appendix A.5.2`, explaining the specific roles of the primary backbone models and the generalization variants. We also corrected citations and clarified data sourcing for specific baseline results in this section.
6. Clarified methodological details regarding the embedding extraction pipeline (modality-specific pooling) and the inference-time path selection logic. We have updated `Sections 3.2, 3.3, and 3.4` to resolve these ambiguities.
7. Clarified key terminology (e.g., "fine-grained semantics," "conflicts," "heterogeneity") to improve the paper's readability and precision. These definitions have been incorporated into the relevant sections of the manuscript.

---

### Meta-Review · Area_Chair_K4Lf · 2026-01-06

**Summary:**

After carefully reviewing the manuscript, all reviewer comments, the authors' comprehensive rebuttal, and the subsequent discussion, I recommend acceptance of this paper. While the initial review scores were mixed (8, 4, 2, 2), the authors have addressed the reviewers' concerns; as the authors point out (and based on my own reading of the reviews), two reviewers based their feedback on misconceptions about the paper (which the authors clarified).

The paper presents TableDART, a novel and practical framework for table understanding that addresses real limitations in existing approaches. The core contribution -- dynamic adaptive routing among text-only, image-only, and fusion paths using a lightweight 2.59M-parameter gating network -- is reasonable and empirically validated. The framework achieves state-of-the-art results across seven benchmarks (average 74.86% accuracy), representing a substantial 4.02% improvement over strong baselines while maintaining computational efficiency. The modular, model-agnostic design allows practitioners to integrate different expert models without prohibitive fine-tuning costs, making it genuinely practical for real-world deployment.

**Reviewer Concerns:**

Reviewer YExZ provided a positive score (8) and explicitly confirmed satisfaction with the authors' responses, maintaining their recommendation for acceptance. Reviewer nQLA initially scored 4 but, after reviewing the comprehensive rebuttal that addressed all conceptual clarifications, additional baselines (TabPedia, SynTab-LLaVA, Qwen2.5-VL), and ablation studies, explicitly committed to providing a positive rating, stating, "My previous concerns have been adequately addressed."

However, Reviewers Z7NP and P1qR (both scoring 2) either did not engage in discussion or maintained their score based on misunderstandings, despite the authors' extensive efforts: adding multiple state-of-the-art baselines, conducting new ablation studies with Qwen2.5-VL demonstrating +7.14% gains, providing detailed efficiency analyses with new figures, and clarifying every technical point!

**Reviewer Scores:**

I do not think the two negative, and possibly biased reviewers would have changed their scores.

---

### Decision · Program_Chairs · 2026-01-26

Accept (Poster)